# A zebrafish embryo screen utilizing gastrulation identifies the HTR2C inhibitor pizotifen as a suppressor of EMT-mediated metastasis

Joji Nakayama[1,2,3,4]*, Lora Tan[1], Yan Li[1], Boon Cher Goh[2], Shu Wang[1,5], Hideki Makinoshima[3,6], Zhiyuan Gong[1]*

[1]Department of Biological Science, National University of Singapore, Singapore, Singapore; [2]Cancer Science Institute of Singapore, National University of Singapore, Singapore, Singapore; [3]Tsuruoka Metabolomics Laboratory, National Cancer Center, Tsuruoka, Japan; [4]Shonai Regional Industry Promotion Center, Tsuruoka, Japan; [5]Institute of Bioengineering and Nanotechnology, Singapore, Singapore; [6]Division of Translational Research, Exploratory Oncology Research and Clinical Trial Center, National Cancer Center, Kashiwa, Japan

**Abstract** Metastasis is responsible for approximately 90% of cancer-associated mortality but few models exist that allow for rapid and effective screening of anti-metastasis drugs. Current mouse models of metastasis are too expensive and time consuming to use for rapid and high-throughput screening. Therefore, we created a unique screening concept utilizing conserved mechanisms between zebrafish gastrulation and cancer metastasis for identification of potential anti-metastatic drugs. We hypothesized that small chemicals that interrupt zebrafish gastrulation might also suppress metastatic progression of cancer cells and developed a phenotype-based chemical screen to test the hypothesis. The screen used epiboly, the first morphogenetic movement in gastrulation, as a marker and enabled 100 chemicals to be tested in 5 hr. The screen tested 1280 FDA-approved drugs and identified pizotifen, an antagonist for serotonin receptor 2C (HTR2C) as an epiboly-interrupting drug. Pharmacological and genetic inhibition of HTR2C suppressed metastatic progression in a mouse model. Blocking HTR2C with pizotifen restored epithelial properties to metastatic cells through inhibition of Wnt signaling. In contrast, HTR2C induced epithelial-to-mesenchymal transition through activation of Wnt signaling and promoted metastatic dissemination of human cancer cells in a zebrafish xenotransplantation model. Taken together, our concept offers a novel platform for discovery of anti-metastasis drugs.

**\*For correspondence:**
zmetastasis@gmail.com (JN);
dbsgzy@nus.edu.sg (ZG)

**Competing interest:** The authors declare that no competing interests exist.

## Editor's evaluation

We are so impressed with this new and ambitious concept for chemical screening using zebrafish embryos to find a novel anti-metastasis drug, Pizotifen. We hope many researchers will use this screening system for anti-cancer drug discovery.

## Introduction

Metastasis, a leading contributor to the morbidity of cancer patients, occurs through multiple steps: invasion, intravasation, extravasation, colonization, and metastatic tumor formation (*Nguyen et al., 2009*; *Welch and Hurst, 2019*; *Chaffer and Weinberg, 2011*). The physical translocation of cancer

cells is an initial step of metastasis and molecular mechanisms of it involve cell motility, the break-down of local basement membrane, loss of cell polarity, acquisition of stem cell-like properties, and epithelial-to-mesenchymal transition (EMT) (*Tsai and Yang, 2013*; *Lu and Kang, 2019*). These cell-biological phenomena are also observed during vertebrate gastrulation in that evolutionarily conserved morphogenetic movements of epiboly, internalization, convergence, and extension progress (*Solnica-Krezel, 2005*). In zebrafish, the first morphogenetic movement, epiboly, is initiated at approximately 4 hr post fertilization (hpf) to move cells from the animal pole to eventually engulf the entire yolk cell by 10 hpf (*Latimer and Jessen, 2010*; *Solnica-Krezel, 2006*). The embryonic cell movements are governed by the molecular mechanisms that are partially shared in metastatic cell dissemination.

At least 50 common genes were shown to be involved in both metastasis and gastrulation progression: Knockdown of these genes in *Xenopus* or zebrafish induced gastrulation defects; conversely, overexpression of these genes conferred metastatic potential on cancer cells and knockdown of these genes suppressed metastasis (*Yang and Weinberg, 2008*; *Dongre and Weinberg, 2019*; *Thiery et al., 2009*; *Nieto et al., 2016*; *Table 1*). This evidence led us to hypothesize that small molecules that interrupt zebrafish gastrulation may suppress metastatic progression of human cancer cells.

Here, we report a unique screening concept based on the hypothesis. Pizotifen, an antagonist for HTR2C, was identified from the screen as a 'hit' that interrupted zebrafish gastrulation. A mouse model of metastasis confirmed pharmacological and genetic inhibition of HTR2C suppressed metastatic progression. Moreover, HTR2C induced EMT and promoted metastatic dissemination of non-metastatic cancer cells in a zebrafish xenotransplantation model. These results demonstrated that this concept could offer a novel high-throughput platform for discovery of anti-metastasis drugs and can be converted to a chemical genetic screening platform.

## Results

### Small molecules interrupting epiboly of zebrafish have a potential to suppress metastatic progression of human cancer cells

Before performing a screening assay, we validated a core of our concept through comparing the genes expressed in zebrafish gastrulation with the genes which expressed in EMT-mediated metastasis. Gene set enrichment analysis (GSEA) demonstrated that 50%-epiboly, shield, and 75%-epiboly stage of zebrafish embryos expressed the genes which promote EMT-mediated metastasis: EMT induction, TGF-β signaling, wnt/β-catenin signaling, Notch signaling (*Figure 1—figure supplement 1*).

We further conducted preliminary experiments to test the hypothesis. First, we examined whether hindering the molecular function of reported genes, whose knockdown induced gastrulation defects in zebrafish, might suppress cell motility and invasion of cancer cells. We chose protein arginine methyltransferase 1 (PRMT1) and cytochrome P450 family 11 (CYP11A1), both of whose knockdown induced gastrulation defects in zebrafish but whose involvement in metastatic progression is unclear (*Tsai et al., 2011*; *Hsu et al., 2006*). Elevated expression of PRMT1 and CYP11A1 was observed in highly metastatic human breast cancer cell lines and knockdown of these genes through RNA interference suppressed the motility and invasion of MDA-MB-231 cells without affecting their viability (*Figure 1—figure supplement 2A-C*).

Next, we conducted an inverse examination of whether chemicals which were reported to suppress metastatic dissemination of cancer cells could interrupt epiboly progression of zebrafish embryos. Niclosamide and vinpocetine are reported to suppress metastatic progression (*Weinbach and Garbus, 1969*; *Sack et al., 2011*; *Huang et al., 2012*; *Szilágyi et al., 2005*). Either niclosamide- or vinpocetine-treated zebrafish embryos showed complete arrest at very early stages or severe delay in epiboly progression, respectively (*Figure 1—figure supplement 2D*).

These results suggest that epiboly could serve as a marker for this screening assay and epiboly-interrupting drugs that are identified through this screening could have the potential to suppress metastatic progression of human cancer cells.

**Table 1.** A list of the genes that are involved between gastrulation and metastasis progression.

A list of the 50 genes that play essential role in governing both metastasis and gastrulation progression. The gastrulation defects in *Xenopus* or zebrafish that are induced by knockdown of each of these genes were indicated. The molecular mechanism in metastasis that is inhibited by knockdown of each of the same genes was indicated.

| Genes | Gastrulation defects | Ref | Effects in metastasis | Ref |
|---|---|---|---|---|
| BMP | Convergence and extension | Kondo, 2007 | EMT | Katsuno et al., 2008 |
| WNT | Convergence and extension | Tada and Smith, 2000 | Migration and invasion | Vincan and Barker, 2008 |
| FGF | Convergence and extension | Yang et al., 2002 | Invision | Nomura et al., 2008 |
| EGF | Epiboly | Song et al., 2013 | Migration | Lu et al., 2001 |
| PDGF | Convergence and extension | Damm and Winklbauer, 2011 | EMT | Jechlinger et al., 2006 |
| CXCL12 | Migration of endodermal cells | Mizoguchi et al., 2008 | Migration and invasion | Shen et al., 2013 |
| CXCR4 | Migration of endodermal cells | Mizoguchi et al., 2008 | Migration and invasion | Shen et al., 2013 |
| PIK3CA | Convergence and extension | Montero et al., 2003 | Migration and invasion | Wander et al., 2013 |
| YES | Epiboly | Tsai et al., 2005 | Migration | Barraclough et al., 2007 |
| FYN | Epiboly | Sharma et al., 2005 | Migration and invasion | Yadav and Denning, 2011 |
| MAPK1 | Epiboly | Krens et al., 2008 | Migration | Radtke et al., 2013 |
| SHP2 | Convergence and extension | Jopling et al., 2007 | Migration | Wang et al., 2005 |
| SNAI1 | Convergence and extension | Ip and Gridley, 2002 | EMT | Batlle et al., 2000 |
| SNAI2 | Mesoderm and neural crest formation | Shi et al., 2011 | EMT | Medici et al., 2008 |
| TWIST1 | Mesoderm formation | Castanon and Baylies, 2002 | EMT | Yang et al., 2004 |
| TBXT | Convergence and extension | Tada and Smith, 2000 | EMT | Fernando et al., 2010 |
| ZEB1 | Epiboly | Vannier et al., 2013 | EMT | Spaderna et al., 2008 |
| GSC | Mesodermal patterning | Sander et al., 2007 | EMT | Hartwell et al., 2006 |
| FOXC2 | Unclear, defects in gastrulation | Wilm et al., 2004 | EMT | Mani et al., 2007 |

*Table 1 continued on next page*

*Table 1 continued*

| Genes | Gastrulation defects | Ref | Effects in metastasis | Ref |
|---|---|---|---|---|
| STAT3 | Convergence and extension | Miyagi et al., 2004 | Migration | Abdulghani et al., 2008 |
| POU5F1 | Epiboly | Lachnit et al., 2008 | EMT | Dai et al., 2013 |
| EZH2 | Unclear, defects in gastrulation | O'Carroll et al., 2001 | Invasion | Ren et al., 2012 |
| EHMT2 | Defects in neurogenesis | Lin et al., 2005 | Migration and invasion | Chen et al., 2010 |
| BMI1 | Defects in skeleton formation | van der Lugt et al., 1994 | EMT | Guo et al., 2011 |
| RHOA | Convergence and extension | Zhu et al., 2006 | Migration and invasion | Yoshioka et al., 1999 |
| CDC42 | Convergence and extension | Choi and Han, 2002 | Migration and invasion | Reymond et al., 2012 |
| RAC1 | Convergence and extension | Habas et al., 2003 | Migration and invasion | Vega and Ridley, 2008 |
| ROCK2 | Convergence and extension | Marlow et al., 2002 | Migration and invasion | Itoh et al., 1999 |
| PAR1 | Convergence and extension | Kusakabe and Nishida, 2004 | Migration | Shi et al., 2004 |
| PRKCI | Convergence and extension | Kusakabe and Nishida, 2004 | EMT | Gunaratne et al., 2013 |
| CAP1 | Convergence and extension | Seifert et al., 2009 | Migration | Yamazaki et al., 2009 |
| EZR | Epiboly | Link et al., 2006 | Migration | Khanna et al., 2004 |
| EPCAM | Epiboly | Slanchev et al., 2009 | Migration and invasion | Ni et al., 2012 |
| ITGB1/ ITA5 | Mesodermal migration | Skalski et al., 1998 | Migration and invasion | Felding-Habermann, 2003 |
| FN1 | Convergence and extension | Marsden and DeSimone, 2003 | Invasion | Malik et al., 2010 |
| HAS2 | Dorsal migration of lateral cells | Bakkers et al., 2004 | Invasion | Kim et al., 2004 |
| MMP14 | Convergence and extension | Coyle et al., 2008 | Invasion | Perentes et al., 2011 |
| COX1 | Epiboly | Cha et al., 2006 | Invasion | Kundu and Fulton, 2002 |
| PTGES | Convergence and extension | Speirs et al., 2010 | Invasion | Wang and Dubois, 2006 |
| SLC39A6 | Anterior migration | Yamashita et al., 2004 | EMT | Lue et al., 2011 |
| GNA12 /13 | Convergence and extension | Lin et al., 2005 | Migration and invasion | Yagi et al., 2011 |

Table 1 continued

| Genes | Gastrulation defects | Ref | Effects in metastasis | Ref |
|---|---|---|---|---|
| OGT | Epiboly | Webster et al., 2009 | Migration and invasion | Lynch et al., 2012 |
| CCN1 | Cell movement | Latinkic et al., 2003 | Migration and invasion | Lin et al., 2012 |
| TRPM7 | Convergence and extension | Liu et al., 2011 | Migration | Middelbeek et al., 2012 |
| MAPKAPK2 | Epiboly | Holloway et al., 2009 | Migration | Kumar et al., 2010 |
| B4GALT1 | Convergence and extension | Machingo et al., 2006 | Invasion | Zhu et al., 2005 |
| IER2 | Convergence and extension | Hong et al., 2011 | Migration | Neeb et al., 2012 |
| TIP1 | Convergence and extension | Besser et al., 2007 | Migration and invasion | Han et al., 2012 |
| PAK5 | Convergence and extension | Faure et al., 2005 | Migration | Gong et al., 2009 |
| MARCKS | Convergence and extension | Iioka et al., 2004 | Migration and invasion | Rombouts et al., 2013 |

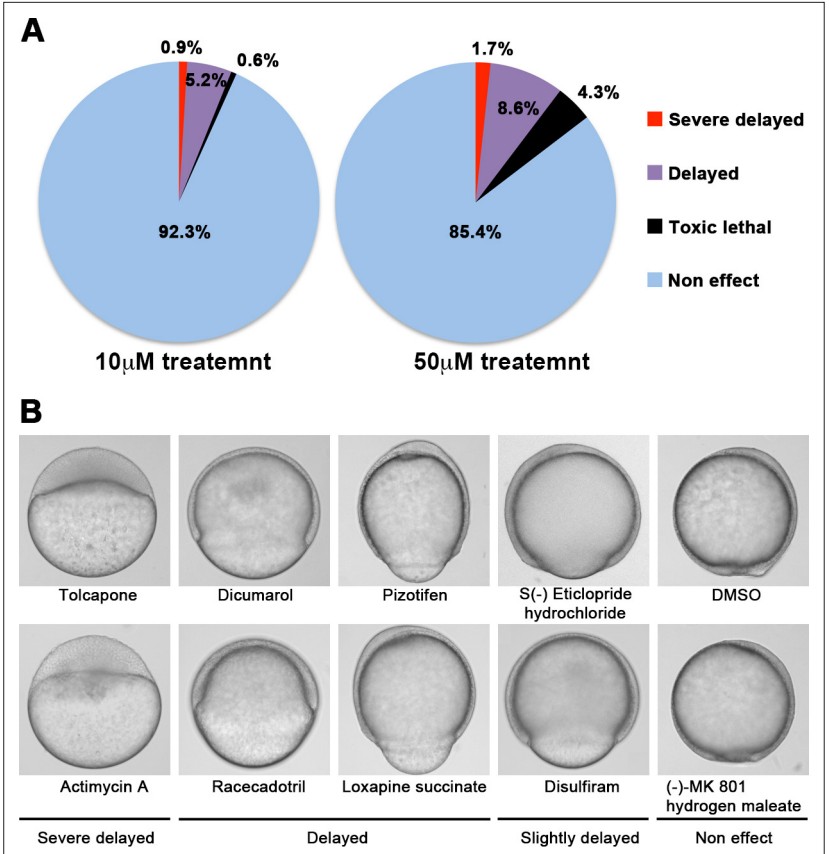

**Figure 1.** A chemical screen for identification of epiboly-interrupting drugs. (**A**) Cumulative results of the chemical screen in which each drug was used at either 10 μM (left) or 50 μM (right) concentrations. 1280 FDA, EMA, or other agencies-approved drugs were subjected to this screening. Positive 'hit' drugs were those that interrupted epiboly progression. (**B**) Representative samples of the embryos that were treated with indicated drugs.

The online version of this article includes the following figure supplement(s) for figure 1:

**Figure supplement 1.** Gene expression profiles obtained from zebrafish embryos at either 50%-epiboly (top left), shield (top right), or 75%-epiboly stage (bottom left) were analyzed based on the hallmark gene sets derived from the Molecular Signatures Database (MSigDB) (*Liberzon et al., 2015*).

**Figure supplement 2.** Epiboly could serve as a marker for this screening.

## 132 FDA-approved drugs induced delayed in epiboly of zebrafish embryos

We screened 1280 FDA, EMA, or other agencies-approved drugs (Prestwick, Inc) in our zebrafish assay. The screening showed that 0.9% (12/1280) of the drugs, including antimycin A and tolcapone, induced severe or complete arrest of embryonic cell movement when embryos were treated with 10 μM. 5.2% (66/1280) of the drugs, such as dicumarol, racecadotril, pizotifen, and S(-)eticlopride hydrochloride, induced either delayed epiboly or interrupted epiboly of the embryos. 93.3% (1194/1280) of drugs have no effect on epiboly progression of the embryos. 0.6% (8/1280) of drugs induced toxic lethality. Epiboly progression was affected more severely when embryos were treated with 50 μM; 1.7% (22/1280) of the drugs induced severe or complete arrest of it. 8.6% (110/1280) of the drugs induced either delayed epiboly or interrupt epiboly of the embryos. 4.3% (55/1280) of drugs induced a toxic lethality (*Figure 1A and B*, *Table 2*). Among the epiboly-interrupting drugs, several drugs have already been reported to inhibit metastasis-related molecular mechanisms: adrenosterone or zardaverine, which target HSD11β1 or PDE3 and -4, respectively, are reported to inhibit EMT (*Nakayama et al., 2020*; *Kolosionek et al., 2009*); racecadotril, which targets enkephalinase, is reported to confer metastatic potential on colon cancer cell (*Sasaki et al., 2014*); and disulfiram, which targets ALDH (aldehyde dehydrogenase), is reported to confer stem-like properties on

**Table 2.** A list of the drugs that interfere with epiboly progression in zebrafish.
Related to *Figure 1*. A list of positive 'hit' drugs that interfered with epiboly progression.
Gastrulation defects or status of each of the zebrafish embryos that were treated with either 10 or
50 µM concentrations are indicated.

| Chemical name | Chemical formula | Effect of 10 µM | Effect of 50 µM |
|---|---|---|---|
| Acitretin | $C_{21}H_{26}O_3$ | Delayed | Delayed |
| Adrenosterone | $C_{19}H_{24}O_3$ | Delayed | Delayed |
| Albendazole | $C_{12}H_{15}N_3O_2S$ | Severe delayed | Severe delayed |
| Alfadolone acetate | $C_{23}H_{34}O_5$ | Delayed | Delayed |
| Alfaxalone | $C_{21}H_{32}O_3$ | Delayed | Delayed |
| Alprostadil | $C_{20}H_{34}O_5$ | Delayed | Delayed |
| Altrenogest | $C_{21}H_{26}O_2$ | Slightly delayed | Delayed |
| Ampiroxicam | $C_{20}H_{21}N_3O_7S$ | Non-effect | Delayed |
| Anethole-trithione | $C_{10}H_8O S_3$ | Delayed | Delayed |
| Antimycin A | $C_{28}H_{40}N_2O_9$ | Delayed | Delayed |
| Avobenzone | $C_{20}H_{22}O_3$ | Delayed | Delayed |
| Benzoxiquine | $C_{16}H_{11}NO_2$ | Non-effect | Delayed |
| Bosentan | $C_{27}H_{29}N_5O_6S$ | Delayed | Delayed |
| Butoconazole nitrate | $C_{19}H_{18}Cl_3N_3O_3S$ | Delayed | Toxic lethal |
| Camptothecine (S,+) | $C_{20}H_{16}N_2O_4$ | Severe delayed | Severe delayed |
| Carbenoxolone disodium salt | $C_{34}H_{48}Na_2O_7$ | Delayed | Toxic lethal |
| Carmofur | $C_{11}H_{16}FN_3O_3$ | Slightly delayed | Delayed |
| Carprofen | $C_{15}H_{12}ClNO_2$ | Severe delayed | Toxic lethal |
| Cefdinir | $C_{14}H_{13}N_5O_5S_2$ | Delayed | Delayed |
| Celecoxib | $C_{17}H_{14}F_3N_3O_2S$ | Delayed | Delayed |
| Chlorambucil | $C_{14}H_{19}Cl_2NO_2$ | Slightly delayed | Delayed |
| Chlorhexidine | $C_{22}H_{30}Cl_2N_{10}$ | Non-effect | Toxic lethal |
| Ciclopirox ethanolamine | $C_{14}H_{24}N_2O_3$ | Delayed | Severe delayed |
| Cinoxacin | $C_{12}H_{10}N_2O_5$ | Delayed | Severe delayed |
| Clofibrate | $C_{12}H_{15}ClO_3$ | Non-effect | Severe delayed |
| Clopidogrel | $C_{16}H_{16}ClNO_2S$ | Non-effect | Delayed |
| Clorgyline hydrochloride | $C_{13}H_{16}Cl_3NO$ | Delayed | Delayed |
| Colchicine | $C_{22}H_{25}NO_6$ | Non-effect | Delayed |
| Deptropine citrate | $C_{29}H_{35}NO_8$ | Delayed | Delayed |
| Desipramine hydrochloride | $C_{18}H_{23}ClN_2$ | Delayed | Delayed |
| Diclofenac sodium | $C_{14}H_{10}Cl_2NNaO_2$ | Delayed | Severe delayed |
| Dicumarol | $C_{19}H_{12}O_6$ | Delayed | Severe delayed |
| Diethylstilbestrol | $C_{18}H_{20}O_2$ | Delayed | Toxic lethal |
| Dimaprit dihydrochloride | $C_6H_{17}Cl_2N_3S$ | Slightly delayed | Delayed |
| Disulfiram | $C_{10}H_{20}N_2S_4$ | Delayed | Delayed |
| Dopamine hydrochloride | $C_8H_{12}ClNO_2$ | Delayed | Delayed |

*Table 2 continued on next page*

*Table 2 continued*

| Chemical name | Chemical formula | Effect of 10 µM | Effect of 50 µM |
|---|---|---|---|
| Eburnamonine (-) | $C_{19}H_{22}N_2O$ | Delayed | Delayed |
| Ethaverine hydrochloride | $C_{24}H_{30}ClNO_4$ | Delayed | Delayed |
| Ethinylestradiol | $C_{20}H_{24}O_2$ | Delayed | Severe delayed |
| Ethopropazine hydrochloride | $C_{19}H_{25}ClN_2S$ | Delayed | Delayed |
| Ethoxyquin | $C_{14}H_{19}NO$ | Non-effect | Delayed |
| Exemestane | $C_{20}H_{24}O_2$ | Slightly delayed | Delayed |
| Ezetimibe | $C_{24}H_{21}F_2NO_3$ | Slightly delayed | Delayed |
| Fenbendazole | $C_{15}H_{13}N_3O_2S$ | Non-effect | Delayed |
| Fenoprofen calcium salt dihydrate | $C_{30}H_{30}CaO_8$ | Slightly delayed | Delayed |
| Fentiazac | $C_{17}H_{12}ClNO_2S$ | Toxic lethal | Toxic lethal |
| Floxuridine | $C_9H_{11}FN_2O_5$ | Delayed | Toxic lethal |
| Flunixin meglumine | $C_{21}H_{28}F_3N_3O_7$ | Delayed | Toxic lethal |
| Flutamide | $C_{11}H_{11}F_3N_2O_3$ | Delayed | Toxic lethal |
| Fluticasone propionate | $C_{25}H_{31}F_3O_5S$ | Non-effect | Delayed |
| Furosemide | $C_{12}H_{11}ClN_2O_5S$ | Delayed | Delayed |
| Gatifloxacin | $C_{19}H_{22}FN_3O_4$ | Delayed | Delayed |
| Gemcitabine | $C_9H_{11}F_2N_3O_4$ | Delayed | Delayed |
| Gemfibrozil | $C_{15}H_{22}O_3$ | Delayed | Toxic lethal |
| Gestrinone | $C_{21}H_{24}O_2$ | Delayed | Delayed |
| Haloprogin | $C_9H_4Cl_3IO$ | Delayed | Toxic lethal |
| Hexachlorophene | $C_{13}H_6Cl_6O_2$ | Delayed | Severe delayed |
| Hexestrol | $C_{18}H_{22}O_2$ | Slightly delayed | Delayed |
| Ibudilast | $C_{14}H_{18}N_2O$ | Non-effect | Delayed |
| Idazoxan hydrochloride | $C_{11}H_{13}ClN_2O_2$ | Slightly delayed | Delayed |
| Idazoxan hydrochloride | $C_{11}H_{13}ClN_2O_2$ | Non-effect | Delayed |
| Idebenone | $C_{19}H_{30}O_5$ | Severe delayed | Toxic lethal |
| Indomethacin | $C_{19}H_{16}ClNO_4$ | Non-effect | Delayed |
| Ipriflavone | $C_{18}H_{16}O_3$ | Delayed | Severe delayed |
| Isotretinoin | $C_{20}H_{28}O_2$ | Non-effect | Severe delayed |
| Isradipine | $C_{19}H_{21}N_3O_5$ | Non-effect | Delayed |
| Lansoprazole | $C_{16}H_{14}F_3N_3O_2S$ | Slightly delayed | Delayed |
| Latanoprost | $C_{26}H_{40}O_5$ | Non-effect | Delayed |
| Leflunomide | $C_{12}H_9F_3N_2O_2$ | Delayed | Severe delayed |
| Letrozole | $C_{17}H_{11}N_5$ | Non-effect | Delayed |
| Lithocholic acid | $C_{24}H_{40}O_3$ | Non-effect | Delayed |
| Lodoxamide | $C_{11}H_6ClN_3O_6$ | Non-effect | Delayed |
| Lofepramine | $C_{26}H_{27}ClN_2O$ | Non-effect | Delayed |
| Loratadine | $C_{22}H_{23}ClN_2O_2$ | Delayed | Delayed |
| Loxapine succinate | $C_{22}H_{24}ClN_3O_5$ | Delayed | Delayed |

*Table 2 continued on next page*

*Table 2 continued*

| Chemical name | Chemical formula | Effect of 10 μM | Effect of 50 μM |
|---|---|---|---|
| Mebendazole | $C_{16}H_{13}N_3O_3$ | Severe delayed | Severe delayed |
| Mebendazole | $C_{22}H_{26}N_2O_2$ | Non-effect | Delayed |
| Meloxicam | $C_{14}H_{13}N_3O_4S_2$ | Delayed | Toxic lethal |
| Methiazole | $C_{12}H_{15}N_3O_2S$ | Delayed | Delayed |
| Mevastatin | $C_{23}H_{34}O_5$ | Non-effect | Delayed |
| MK 801 hydrogen maleate | $C_{20}H_{19}NO_4$ | Slightly delayed | Delayed |
| Nabumetone | $C_{15}H_{16}O_2$ | Non-effect | Severe delayed |
| Naftopidil dihydrochloride | $C_{24}H_{30}Cl_2N_2O_3$ | Slightly delayed | Delayed |
| Nandrolone | $C_{18}H_{26}O_2$ | Delayed | Delayed |
| Naproxen sodium salt | $C_{14}H_{13}NaO_3$ | Delayed | Delayed |
| Niclosamide | $C_{13}H_8Cl_2N_2O_4$ | Delayed | Delayed |
| Nifekalant | $C_{19}H_{27}N_5O_5$ | Delayed | Delayed |
| Niflumic acid | $C_{13}H_9F_3N_2O_2$ | Delayed | Delayed |
| Nimesulide | $C_{13}H_{12}N_2O_5S$ | Non-effect | Delayed |
| Nisoldipine | $C_{20}H_{24}N_2O_6$ | Delayed | Toxic lethal |
| Nitazoxanide | $C_{12}H_9N_3O_5S$ | Severe delayed | Severe delayed |
| Norethindrone | $C_{20}H_{26}O_2$ | Non-effect | Delayed |
| Norgestimate | $C_{23}H_{31}NO_3$ | Slightly delayed | Delayed |
| Oxfendazol | $C_{15}H_{13}N_3O_3S$ | Slightly delayed | Delayed |
| Oxibendazol | $C_{12}H_{15}N_3O_3$ | Severe delayed | Severe delayed |
| Oxymetholone | $C_{21}H_{32}O_3$ | Slightly delayed | Delayed |
| Parbendazole | $C_{13}H_{17}N_3O_2$ | Severe delayed | Severe delayed |
| Parthenolide | $C_{15}H_{20}O_3$ | Non-effect | Delayed |
| Penciclovir | $C_{10}H_{15}N_5O_3$ | Non-effect | Delayed |
| Pentobarbital | $C_{11}H_{18}N_2O_3$ | Non-effect | Delayed |
| Phenazopyridine hydrochloride | $C_{11}H_{12}ClN_5$ | Delayed | Toxic lethal |
| Phenothiazine | $C_{12}H_9NS$ | Non-effect | Delayed |
| Phenoxybenzamine hydrochloride | $C_{18}H_{23}Cl_2NO$ | Non-effect | Delayed |
| Pizotifen malate | $C_{23}H_{27}NO_5S$ | Delayed | Severe delayed |
| Pramoxine hydrochloride | $C_{17}H_{28}ClNO_3$ | Slightly delayed | Delayed |
| Prilocaine hydrochloride | $C_{13}H_{21}ClN_2O$ | Non-effect | Delayed |
| Primidone | $C_{12}H_{14}N_2O_2$ | Slightly delayed | Delayed |
| Racecadotril | $C_{21}H_{23}NO_4S$ | Slightly delayed | Delayed |
| Riluzole hydrochloride | $C_8H_6ClF_3N_2OS$ | Non-effect | Delayed |
| Ritonavir | $C_{37}H_{48}N_6O_5S_2$ | Non-effect | Severe delayed |
| S(-)Eticlopride hydrochloride | $C_{17}H_{26}Cl_2N_2O_3$ | Delayed | Delayed |
| Salmeterol | $C_{25}H_{37}NO_4$ | Non-effect | Delayed |
| Streptomycin sulfate | $C_{42}H_{84}N_{14}O_{36}S_3$ | Non-effect | Delayed |
| Sulconazole nitrate | $C_{18}H_{16}Cl_3N_3O_3S$ | Delayed | Delayed |

*Table 2 continued on next page*

*Table 2 continued*

| Chemical name | Chemical formula | Effect of 10 μM | Effect of 50 μM |
|---|---|---|---|
| Tegafur | $C_8H_9FN_2O_3$ | Delayed | Delayed |
| Telmisartan | $C_{33}H_{30}N_4O_2$ | Severe delayed | Toxic lethal |
| Tenatoprazole | $C_{16}H_{18}N_4O_3S$ | Non-effect | Delayed |
| Terbinafine | $C_{21}H_{25}N$ | Non-effect | Delayed |
| Thimerosal | $C_9H_9HgNaO_2S$ | Non-effect | Delayed |
| Thiorphan | $C_{12}H_{15}NO_3S$ | Delayed | Delayed |
| Tolcapone | $C_{14}H_{11}NO_5$ | Severe delayed | Severe delayed |
| Topotecan | $C_{23}H_{23}N_3O_5$ | Delayed | Delayed |
| Tracazolate hydrochloride | $C_{16}H_{25}ClN_4O_2$ | Severe delayed | Delayed |
| Tribenoside | $C_{29}H_{34}O_6$ | Delayed | Delayed |
| Triclabendazole | $C_{14}H_9Cl_3N_2OS$ | Delayed | Delayed |
| Triclosan | $C_{12}H_7Cl_3O_2$ | Delayed | Severe delayed |
| Trioxsalen | $C_{14}H_{12}O_3$ | Delayed | Delayed |
| Troglitazone | $C_{24}H_{27}NO_5S$ | Severe delayed | Toxic lethal |
| Valproic acid | $C_8H_{16}O_2$ | Non-effect | Delayed |
| Voriconazole | $C_{16}H_{14}F_3N_5O$ | Non-effect | Delayed |
| Zardaverine | $C_{12}H_{10}F_2N_2O_3$ | Slightly delayed | Delayed |
| Zuclopenthixol dihydrochloride | $C_{22}H_{27}Cl_3N_2OS$ | Delayed | Delayed |

metastatic cancer cells (*Liu et al., 2013*). This evidence suggests that epiboly-interrupting drugs have the potential for suppressing metastasis of human cancer cells.

## Identified drugs suppressed cell motility and invasion of human cancer cells

It has been reported that zebrafish have orthologues to 86% of 1318 human drug targets (*Gunnarsson et al., 2008*). However, it was not known whether the epiboly-interrupting drugs could suppress metastatic dissemination of human cancer cells. To test this, we subjected the 78 epiboly-interrupting drugs that showed a suppressor effect on epiboly progression at a 10 μM concentration to in vitro experiments using a human cancer cell line. The experiments examined whether the drugs could suppress cell motility and invasion of MDA-MB-231 cells through a Boyden chamber. Before conducting the experiment, we investigated whether these drugs might affect viability of MDA-MB-231 cells using an MTT assay. Out of the 78 drugs, 16 of them strongly affected cell viability at concentrations less than 1 μM and were not used in the cell motility experiments. The remaining 62 drugs were assayed in Boyden chamber motility experiments. Out of the 62 drugs, 20 of the drugs inhibited cell motility and invasion of MDA-MB-231 cells without affecting cell viability. Among the 20 drugs, hexachlorophene and nitazoxanide were removed since the primary targets of the drugs, D-lactate dehydrogenase and pyruvate ferredoxin oxidoreductase, are not expressed in mammalian cells. With the exception of ipriflavone, whose target is still unclear, the known primary targets of the remaining 17 drugs are reported to be expressed by mammalian cells (*Figure 2A* and *Table 3*).

We confirmed that highly metastatic human cancer cell lines expressed target genes through western blotting analyses. Among the genes, serotonin receptor 2C (HTR2C), which is a primary target of pizotifen, was highly expressed in only metastatic cell lines (*Figure 2B* and *Figure 2—figure supplement 2A*). Clinical data also shows that that HTR2C expression is correlated with tumor stage of breast cancer patients and is higher in metastatic and Her2/neu-overexpressing tumors (*Pai et al., 2009*). Pizotifen suppressed cell motility and invasion of several highly metastatic human cancer cell lines in a dose-dependent manner (*Figure 2C*). Similarly, dopamine receptor D2 (DRD2), which is a

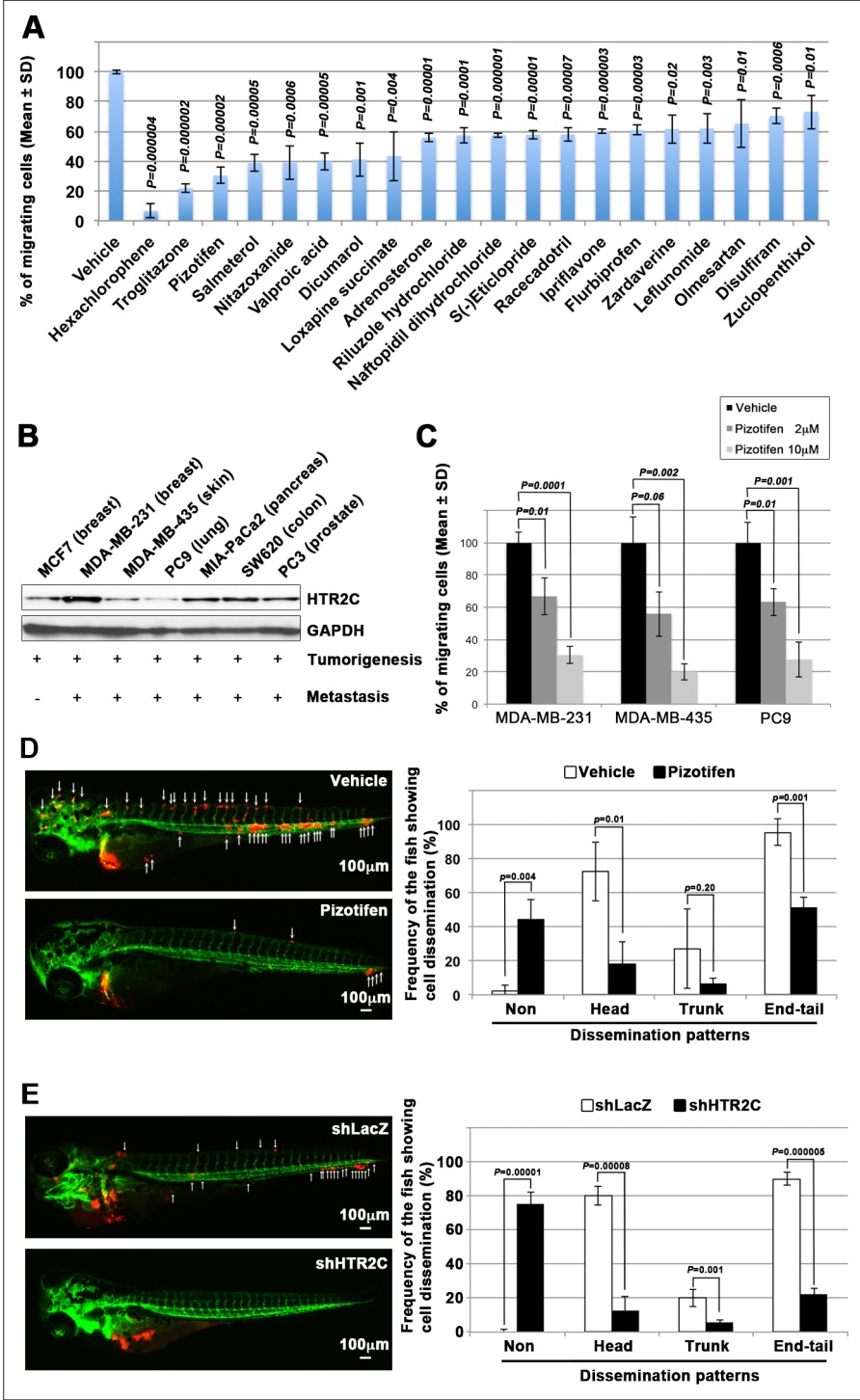

**Figure 2.** Pizotifen, one of epiboly-interrupting drugs, suppressed metastatic dissemination of human cancer cells lines in vivo and vitro. (**A**) Effect of the epiboly-interrupting drugs on cell motility and invasion of MBA-MB-231 cells. MBA-MB-231 cells were treated with vehicle or each of the epiboly-interrupting drugs and then subjected to Boyden chamber assays. Fetal bovine serum (1% v/v) was used as the chemoattractant in both assays. Each experiment was performed at least twice. (**B**) Western blot analysis of HTR2C levels (top) in a non-metastatic human cancer cell line, MCF7 (breast) and highly metastatic human cancer cell lines, MDA-MB-231 (breast), MDA-MB-435 (melanoma), PC9 (lung), MIA-PaCa2 (pancreas), PC3 (prostate), and SW620 (colon); GAPDH loading control is shown (bottom). (**C**) Effect of pizotifen on cell motility and invasion of MBA-MB-231, MDA-MB-435, and PC9 cells. Either vehicle or pizotifen treated the cells were subjected to Boyden chamber assays. Fetal bovine

*Figure 2 continued on next page*

*Figure 2 continued*

serum (1% v/v) was used as the chemoattractant in both assays. Each experiment was performed at least twice. (**D**) and (**E**) Representative images of dissemination of 231R, shLacZ 231R or shHTR2C 231R cells in zebrafish xenotransplantation model. The fish larvae that were inoculated with 231R cells were treated with either vehicle (top left) or the drug (lower left) (**D**). The fish larvae that were inoculated with either shLacZ 231R or shHTR2C 231R cells (lower left) (**E**). White arrows head indicate disseminated 231R cells. The images were shown in 4× magnification. Scale bar, 100 µm. The mean frequencies of the fish showing head, trunk, or end-tail dissemination were counted (graph on right). Each value is indicated as the mean ± SEM of two independent experiments. Statistical analysis was determined by Student's t test.

The online version of this article includes the following figure supplement(s) for figure 2:

**Figure supplement 1.** Blocking Dopamine receptor D2 with S(-) Eticlopride hydrochloride suppressed cell motility and invasion of highly metastatic human cancer cells in a dose-dependent manner.

**Figure supplement 2.** Pizotifen suppressed metastatic dissemination of MDA-MB-231 and MIA-PaCa2 cells in a zebrafish xenotransplantation model.

---

primary target of S(-)eticlopride hydrochloride, was highly expressed in only metastatic cell lines, and the drug suppressed cell motility and invasion of these cells in a dose-dependent manner (***Figure 2—figure supplement 2A-C***).

These results indicate that a number of the epiboly-interrupting drugs also have suppressor effects on cell motility and invasion of highly metastatic human cancer cells.

**Table 3.** Primary targets of the identified drugs.

| The identified drugs | Primary targets of the identified drugs |
|---|---|
| Hexachlorophene | D-Lactate dehydrogenase (D-LDH), not expressed in mammalian cells |
| Troglitazone | Agonist for peroxisome proliferator-activated receptor α and γ (PPARα and -γ) |
| Pizotifen malate | 5-Hydroxytryptamine receptor 2C (HTR2C) |
| Salmeterol | Adrenergic receptor beta 2 (ADRB2) |
| Nitazoxanide | Pyruvate ferredoxin oxidoreductase (PFOR), not expressed in mammalian cells |
| Valproic acid | Histone deacetylases (HDACs) |
| Dicumarol | NAD(P)H dehydrogenase quinone 1 (NQO1) |
| Loxapine succinate | Dopamine receptor D2 and D4 (DRD2 and DRD4) |
| Adrenosterone | Hydroxysteroid (11-beta) dehydrogenase 1 (HSD11β1) |
| Riluzole hydrochloride | Glutamate R and voltage-dependent Na+ channel |
| Naftopidil dihydrochloride | 5-Hydroxytryptamine receptor 1A (HTR1A) and α1-adrenergic receptor (AR) |
| S(-)Eticlopride hydrochloride | Dopamine receptor D2 (DRD2) |
| Racecadotril | Membrane metallo-endopeptidase (MME) |
| Ipriflavone | Unknown |
| Flurbiprofen | Cyclooxygenase 1 and 2 (Cox1 and -2) |
| Zardaverine | Phosphodiesterase III/IV (PDE3/4) |
| Leflunomide | Dihydroorotate dehydrogenase (DHODH) |
| Olmesartan | Angiotensin II receptor alpha |
| Disulfiram | Aldehyde dehydrogenase (ALDH) Dopamine β-hydroxylase (DBH) |
| Zuclopenthixol dihydrochloride | Dopamine receptors D1 and D2 (DRD1 and -2) |

## Pizotifen suppressed metastatic dissemination of human cancer cells in a zebrafish xenotransplantation model

While a number of the epiboly-interrupting drugs suppressed cell motility and invasion of human cell lines in vitro, it was still unclear whether the drugs could suppress metastatic dissemination of cancer cells in vivo. Therefore, we examined whether the identified drugs could suppress metastatic dissemination of these human cancer cells in a zebrafish xenotransplantation model. Pizotifen was selected to test since HTR2C was overexpressed only in highly metastatic cell lines supporting the hypothesis that it could be a novel target for blocking metastatic dissemination of cancer cells (*Figure 2B*). Red fluorescent protein (RFP)-labelled MDA-MB-231 (231R) cells were injected into the duct of Cuvier of *Tg* (*kdrl:eGFP*) zebrafish at 2 dpf and then maintained in the presence of either vehicle or pizotifen. Twenty-four hours post injection, the numbers of fish showing metastatic dissemination of 231R cells were measured via fluorescence microscopy. In this model, the dissemination patterns were generally divided into three categories: (i) head dissemination, in which disseminated 231R cells exist in the vessel of the head part; (ii) trunk dissemination, in which the cells were observed in the vessel dilating from the trunk to the tail; (iii) end-tail dissemination, in which the cells were observed in the vessel of the end-tail part (*Nakayama et al., 2020*).

Three independent experiments revealed that the frequencies of fish in the drug-treated group showing head, trunk, or end-tail dissemination significantly decreased to 55.3% ± 7.5%, 28.5 ± 5.0%, or 43.5% ± 19.1% when compared with those in the vehicle-treated group; 95.8% ± 5.8%, 47.1 ± 7.7%, or 82.6% ± 12.7%. Conversely, the frequency of the fish in the drug-treated group not showing any dissemination significantly increased to 45.4% ± 0.5% when compared with those in the vehicle-treated group; 2.0% ± 2.9% (*Figure 2D*, *Figure 2—figure supplement 2* and *Table 4*).

Similar effects were observed in another xenograft experiments using an RFP-labelled human pancreatic cancer cell line, MIA-PaCa-2 (MP2R). In the drug-treated group, the frequencies of the fish showing head, trunk, or end-tail dissemination significantly decreased to 15.3% ± 6.7%, 6.2% ± 1.3%, or 41.1% ± 1.5%; conversely, the frequency of the fish not showing any dissemination significantly increased to 46.3% ± 8.9% when compared with those in the vehicle-treated group; 74.5% ± 11.1%, 18.9% ± 14.9%, 77.0% ± 9.0%, or 17.2% ± 0.7% (*Figure 2—figure supplement 2A* and *Table 5*).

To eliminate the possibility that the metastasis suppressing effects of pizotifen might result from off-target effects of the drug, we conducted validation experiments to determine whether knockdown of HTR2C would show the same effects. Sub-clones of 231R cells that expressed short hairpin RNA (shRNA) targeting either LacZ or HTR2C were injected into the fish at 2 dpf and the fish were maintained in the absence of drug. In the fish that were inoculated with shHTR2C 231R cells, the frequencies of the fish showing head, trunk, and end-tail dissemination significantly decreased to 6.7% ± 4.9%, 6.7% ± 0.7%, or 20.0% ± 16.5%; conversely, the frequency of the fish not showing any dissemination significantly increased to 80.0% ± 4.4% when compared with those that were inoculated with shLacZ 231R cells; 80.0% ± 27.1%, 20.0% ± 4.5%, 90.0% ± 7.7%, or 0% (*Figure 2E* and *Table 6*).

These results indicate that pharmacological and genetic inhibition of HTR2C suppressed metastatic dissemination of human cancer cells in vivo.

## Pizotifen suppressed metastasis progression of a mouse model of metastasis

We examined the metastasis-suppressor effect of pizotifen in a mouse model of metastasis (*Tao et al., 2008*). Luciferase-expressing 4T1 murine mammary carcinoma cells were inoculated into the mammary fat pads (MFP) of female BALB/c mice. On day 2 post inoculation, the mice were randomly assigned to two groups and one group received once daily intraperitoneal injections of 10 mg/kg pizotifen while the other group received a vehicle injection. Bioluminescence imaging and tumor measurement revealed that the sizes of the primary tumors in pizotifen-treated mice were equal to those in the vehicle-treated mice on day 10 post inoculation. The primary tumors were resected after the analyses. Immunofluorescence (IF) staining also demonstrated that the percentage of Ki67-positive cells in the resected primary tumors of pizotifen-treated mice were the same as those of vehicle-treated mice (*Figure 3A–C*), additionally, both groups showed less than 1% cleaved caspase 3 positive cells (*Figure 3—figure supplement 1*). Therefore, no anti-tumor effect of pizotifen was observed on the primary tumor. After 70 days from inoculation, bioluminescence imaging detected light emitted in the lungs, livers, and lymph nodes of vehicle-treated mice but not those of pizotifen-treated mice

**Table 4.** Effects of pharmacological inhibition of HTR2C on metastatic dissemination of MDA-MB-231 cells in zebrafish xenografted models. Related to *Figure 2D*. The numbers and frequencies of the fish showing the dissemination patterns in vehicle- or pizotifen-treated group were indicated. The fish showed both patterns of dissemination were redundantly counted in this analysis.

| | | Experiment #1 | Experiment #2 | Experiment #3 | Average of experiments |
|---|---|---|---|---|---|
| Drug: Vehicle Cell: MDA-MB-231 | Non-dissemination | 0% n1 = 0/17 | 0% n2 = 0/12 | 6.66% n3 = 1/15 | 2.22% ± 3.84% |
| | Head | 58.82% n1 = 10/17 | 91.66% n2 = 11/12 | 6.66% n3 = 1/15 | 72.38% ± 17.15% |
| | Trunk | 52.94% n1 = 9/17 | 8.33% n2 = 1/12 | 20% n3 = 2/15 | 27.09% ± 23.13% |
| | End-tail | 100% n1 = 17/17 | 100% n2 = 12/12 | 86.66% n3 = 13/15 | 95.55% ± 7.69% |
| Drug: Pizotifen Cell: MDA-MB-231 | Non-dissemination | 55% n1 = 11/20 | 31.57% n2 = 6/19 | 45.45 % n3 = 10/22 | 44.01% ± 11.77% |
| | Head | 5% n1 = 1/20 | 31.57% n2 = 6/19 | 18.18% n3 = 4/22 | 18.25% ± 13.28% |
| | Trunk | 5% n1 = 1/20 | 10.52% n2 = 2/19 | 4.45% n3 = 1/22 | 6.69% ± 3.32% |
| | End-tail | 45% n1 = 9/20 | 57.89% n2 = 11/19 | 50% n3 = 11/22 | 50.96% ± 6.50% |

**Table 5.** Effects of pharmacological inhibition of HTR2C on metastatic dissemination of Mia-PaCa2 cells in zebrafish xenografted models.

Related to *Figure 4*. The numbers and frequencies of the fish showing the dissemination patterns in vehicle- or pizotifen-treated group were indicated. The fish showed both patterns of dissemination were redundantly counted in this analysis.

| | | Experiment _#1 | Experiment _#2 | Average of experiments |
|---|---|---|---|---|
| Drug: Vehicle Cell: MIA-PaCa2 | Non-dissemination | 17.64% n1 = 3/17 | 16.66% n2 = 2/12 | 17.15% + 0.69% |
| | Head | 82.35% n1 = 14/17 | 66.66% n2 = 8/12 | 74.50% + 11.09% |
| | Trunk | 29.41% n1 = 5/17 | 8.33% n2 = 1/12 | 18.87% + 14.90% |
| | End-tail | 70.58% n1 = 12/17 | 83.33% n2 = 10/17 | 76.96% + 9.01 |
| Drug: Pizotifen Cell: MIA-PaCa2 | Non-dissemination | 40% n1 = 4/10 | 52.63% n2 = 10/19 | 46.31% + 8.93% |
| | Head | 20% n1 = 2/10 | 10.52% n2 = 2/19 | 15.26% + 6.69% |
| | Trunk | 10% n1 = 1/10 | 5.26% n2 = 1/19 | 7.63% + 3.34% |
| | End-tail | 40% n1 = 4/10 | 42.05% n2 = 8/19 | 41.4% + 1.48% |

(*Figure 3C*). Vehicle-treated mice formed 5–50 metastatic nodules per lung in all 10 mice analyzed; conversely, pizotifen-treated mice (n = 10) formed 0–5 nodules per lung in all 10 mice analyzed (*Figure 3D*). Histological analyses confirmed that metastatic lesions in the lungs were detected in all vehicle-treated mice; conversely, they were detected in only 2 of 10 pizotifen-treated mice and the rest of the mice showed metastatic colony formations around the bronchiole of the lung. In addition, 4 of 10 vehicle-treated mice exhibited metastasis in the liver and the rest showed metastatic colony formation around the portal tract of the liver. In contrast, none of 10 pizotifen-treated mice showed liver metastases and only half of the 10 mice showed metastatic colony formation around the portal tract (*Figure 3E*). These results indicate that pizotifen can suppress metastasis progression without affecting primary tumor growth.

To eliminate the possibility that the metastasis suppressing effects of pizotifen might result from off-target effects, we conducted validation experiments to determine whether knockdown of HTR2C would show the same effects. The basic experimental process followed the experimental design described above except that sub-clones of 4T1 cells that expressed shRNA targeting either LacZ or HTR2C were injected into the MFP of female BALB/c mice and the mice were maintained without drug. Histological analyses revealed that all of the mice (n = 5) that were inoculated with 4T1 cells

**Table 6.** Effects of genetic inhibition of HTR2C on metastatic dissemination of MDA-MB-231 cells in zebrafish xenografted models.

Related to *Figure 2E*. The numbers and frequencies of the fish showing the dissemination patterns in the zebrafish that were inoculated with either shLacZ or shHTR2C MDA-MB-231 cells were indicated. The fish showed both patterns of dissemination were redundantly counted in this analysis.

| | | Experiment _#1 | Experiment _#2 | Average of experiments |
|---|---|---|---|---|
| shLacZ | Non-dissemination | 0% n1 = 0/10 | 0% n2 = 0/10 | 0% |
| | Head | 60% n1 = 6/10 | 100% n2 = 10/10 | 80% ± 28.28% |
| | Trunk | 30% n1 = 3/10 | 10% n2 = 1/10 | 20% ± 14.14% |
| | End-tail | 80% n1 = 8/10 | 100% n2 = 10/10 | 90% ± 14.14 |
| shHTR2C | Non-dissemination | 80% n1 = 12/15 | 76.84% n2 = 14/19 | 76.84 ± 4.46% |
| | Head | 6.66% n1 = 1/15 | 15.78% n2 = 3/19 | 11.22% ± 6.45% |
| | Trunk | 6.66% n1 = 1/15 | 5.26% n2 = 1/19 | 5.96% ± 0.99% |
| | End-tail | 20% n1 = 3/15 | 26.31% n2 = 5/19 | 23.15% ± 4.46% |

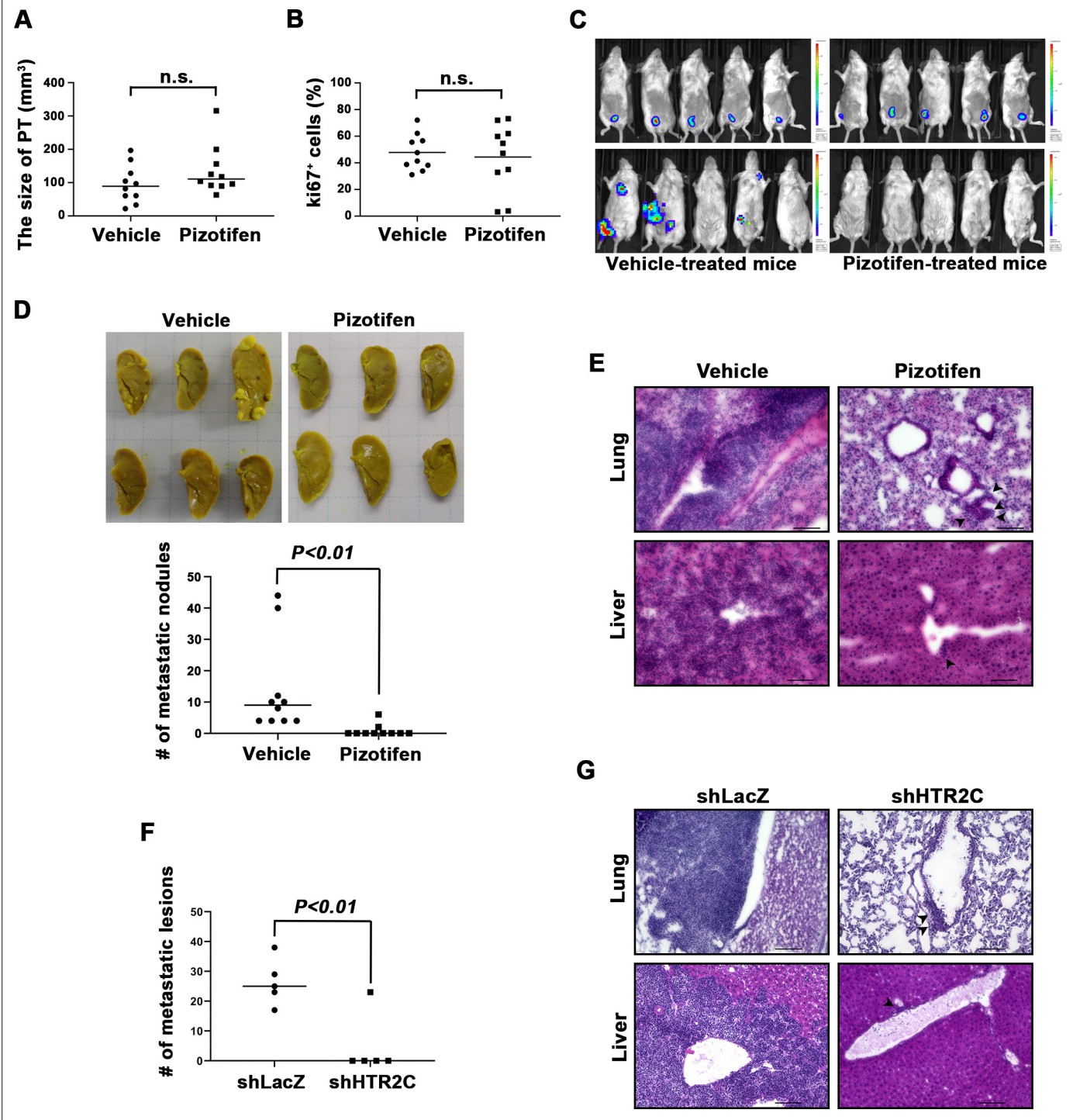

**Figure 3.** Pizotifen suppressed metastatic progression in a mouse model of metastasis. (**A**) Mean volumes (n = 10 per group) of 4T1 primary tumors formed in the mammary fat pad of either vehicle- or pizotifen-treated mice at day 10 post injection. (**B**) Ki67 expression level in 4T1 primary tumors formed in the mammary fat pad of either vehicle- or pizotifen-treated mice at day 10 post injection. The mean expression levels of Ki67 (n = 10 mice per group) were determined and were calculated as the mean ration of Ki67-positive cells to 4',6-diamidino-2-phenylindole (DAPI) area. (**C**) Representative images of primary tumors on day 10 post injection (top panels) and metastatic burden on day 70 post injection (bottom panels) taken using an IVIS Imaging System. (**D**) Representative images of the lungs from either vehicle- (top) or pizotifen-treated mice (bottom) at 70 days post tumor inoculation. Number of metastatic nodules in the lung of either vehicle- or pizotifen-treated mice (right). (**E**) Representative hematoxylin and eosin (H&E) staining of the lung (top) and liver (bottom) from either vehicle- or pizotifen-treated mice. Black arrow heads indicate metastatic 4T1 cells. (**F**) The mean number of metastatic lesions in step sections of the lungs from the mice that were inoculated with 4T1-12B cells expressing short hairpin RNA (shRNA) targeting

*Figure 3 continued on next page*

*Figure 3 continued*

for either LacZ or HTR2C. (**G**) Representative H&E staining of the lung and liver from the mice that were inoculated with 4T1-12B cells expressing shRNA targeting for either LacZ or HTR2C. Black arrow heads indicate metastatic 4T1 cells. Each value is indicated as the mean ± SEM. Statistical analysis was determined by Student's t test.

The online version of this article includes the following figure supplement(s) for figure 3:

**Figure supplement 1.** Cleaved caspase 3 expression level in 4T1 primary tumors formed in the mammary fat pad of either vehicle- or pizotifen-treated mice at day 10 post injection.

expressing shRNA targeting LacZ showed metastases in the lungs. The mean number of metastatic lesions in a lung was 26.4 ± 7.8. In contrast, only one of the mice (n = 5) were inoculated with 4T1 cells expressing shRNA targeting HTR2C showed metastases in the lungs and the rest of the mice showed metastatic colony formation around the bronchiole of the lung. The mean number of metastatic lesions in the lung significantly decreased to 10% of those of mice that were inoculated with 4T1 cells expressing shRNA targeting LacZ (*Figure 3F–H*).

Taken together, pharmacological and genetic inhibition of HTR2C showed an anti-metastatic effect in the 4T1 model system.

## HTR2C promoted EMT-mediated metastatic dissemination of human cancer cells

Although pharmacological and genetic inhibition of HTR2C inhibited metastasis progression, a role for HTR2C on metastatic progression has not been reported. Therefore, we examined whether HTR2C could confer metastatic properties on poorly metastatic cells.

First, we established a stable sub-clone of MCF7 human breast cancer cells expressing either vector control or HTR2C. Vector control expressing MCF7 cells maintained highly organized cell-cell adhesion and cell polarity; however, HTR2C-expressing MCF7 cells led to loss of cell-cell contact and cell scattering. The cobblestone-like appearance of these cells was replaced by a spindle-like, fibroblastic morphology. Western blotting and IF analyses revealed that HTR2C-expressing MCF7 cells showed loss of E-cadherin and EpCAM, and elevated expressions of N-cadherin, vimentin, and an EMT-inducible transcriptional factor ZEB1. Similar effects were validated through another experiment using an immortal keratinocyte cell line, HaCaT cells, in that HTR2C-expressing HaCaT cells also showed loss of cell-cell contact and cell scattering with loss of epithelial markers and gain of mesenchymal markers (*Figure 4A–C* and *Figure 4—figure supplement 1A*). Therefore, both the morphological and molecular changes in the HTR2C-expressing MCF7 and HaCaT cells demonstrated that these cells had undergone an EMT.

Next, we examined whether HTR2C-driven EMT could promote metastatic dissemination of human cancer cells. Boyden chamber assay revealed that HTR2C expressing MCF7 cells showed an increased cell motility and invasion compared with vector control-expressing MCF7 cells in vitro (*Figure 4D*). Moreover, we conducted in vivo examination of whether HTR2C expression could promote metastatic dissemination of human cancer cells in a zebrafish xenotransplantation model. RFP-labelled MCF7 cells expressing either vector control or HTR2C were injected into the duct of Cuvier of *Tg* (*kdrl:eGFP*) zebrafish at 2 dpf. Twenty-four hours post injection, the frequencies of the fish showing metastatic dissemination of the inoculated cells were measured using fluorescence microscopy. In the fish that were inoculated with HTR2C expressing MCF7 cells, the frequencies of the fish showing head, trunk, and end-tail dissemination significantly increased to 96.7% ± 4.7%, 68.8% ± 6.4%, or 89.5% ± 3.4%; conversely, the frequency of the fish not showing any dissemination decreased to 0% when compared with those in the fish that were inoculated with vector control expressing MCF7 cells; 33.1% ± 18.5%, 0%, 56.9% ± 4.4%, or 43% (*Figure 4E*, *Figure 4—figure supplement 1B* and *Table 7*).

These results indicated that HTR2C promoted metastatic dissemination of cancer cells through induction of EMT, and suggest that the screen can easily be converted to a chemical genetic screening platform.

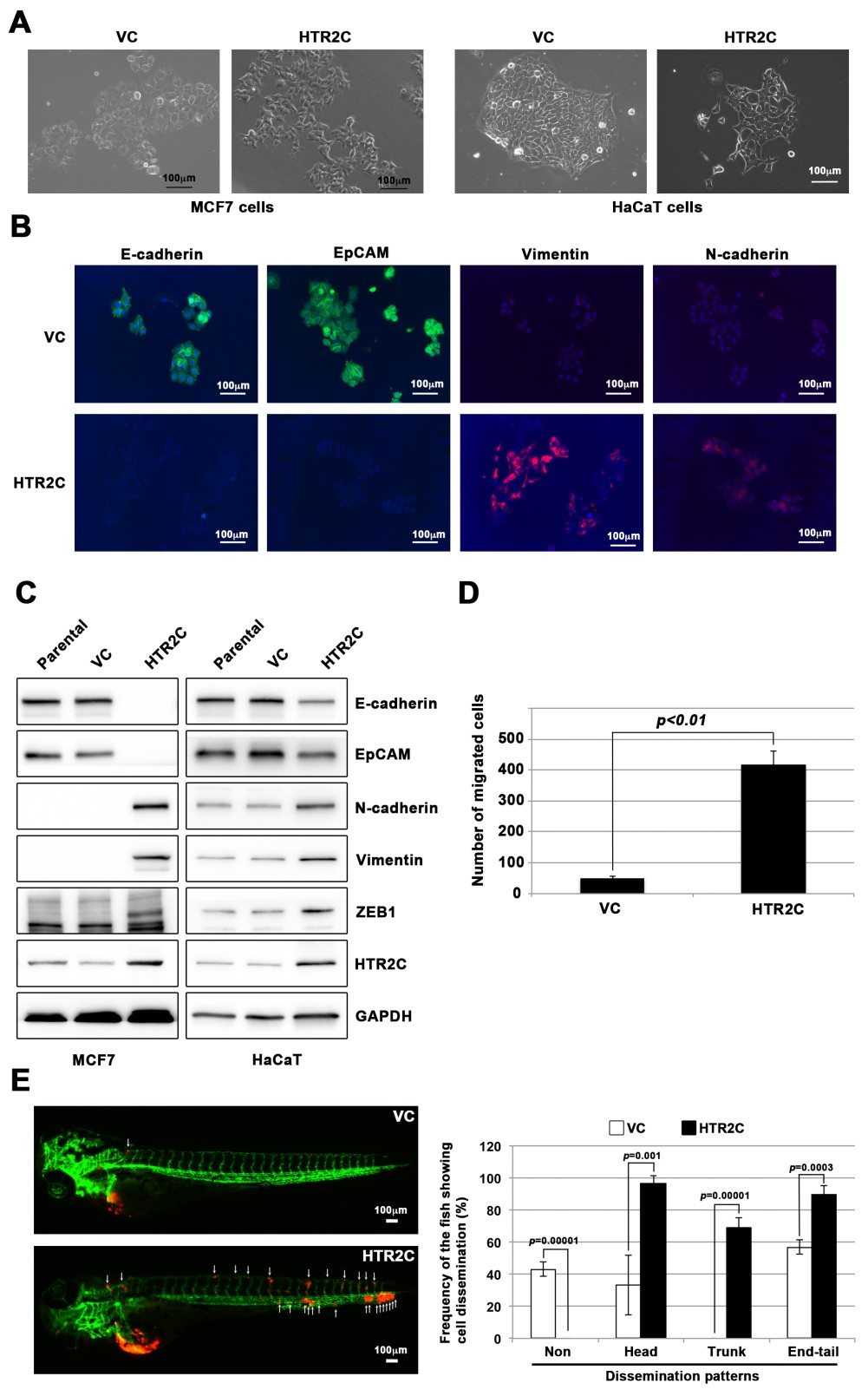

**Figure 4.** HTR2C induced epithelial-to-mesenchymal transition (EMT)-mediated metastatic dissemination of human cancer cells. (**A**) The morphologies of the MCF7 and HaCaT cells expressing either the control vector or HTR2C were revealed by phase contrast microscopy. (**B**) Immunofluorescence staining of E-cadherin, EpCAM, vimentin, and N-cadherin expressions in the MCF7 cells from A. (**C**) Expression of E-cadherin, EpCAM, vimentin,

*Figure 4 continued on next page*

*Figure 4 continued*

N-cadherin, and HTR2C was examined by western blotting in the MCF7 and HaCaT cells; GAPDH loading control is shown (bottom). (**D**) Effect of HTR2C on cell motility and invasion of MCF7 cells. MCF7 cells were subjected to Boyden chamber assays. Fetal bovine serum (1% v/v) was used as the chemoattractant in both assays. Each experiment was performed at least twice. (**E**) Representative images of dissemination patterns of MCF7 cells expressing either the control vector (top left) or HTR2C (lower left) in a zebrafish xenotransplantation model. White arrow heads indicate disseminated MCF7 cells. The mean frequencies of the fish showing head, trunk, or end-tail dissemination tabulated (right). Each value is indicated as the mean ± SEM of two independent experiments. Statistical analysis was determined by Student's t test.

The online version of this article includes the following figure supplement(s) for figure 4:

**Figure supplement 1.** HTR2C promoted EMT-mediated metastatic dissemination of poorly metastatic human cancer cells in a zebrafish xenotransplantation model.

## Pizotifen induced mesenchymal-to-epithelial transition through inhibition of Wnt signaling

Finally, we elucidated the mechanism of action of how pizotifen suppressed metastasis, especially metastatic dissemination of cancer cells. Our results showed that HTR2C induced EMT and that pharmacological and genetic inhibition of HTR2C suppressed metastatic dissemination of MDA-MB-231 cells that had already transitioned to mesenchymal-like traits via EMT. Therefore, we speculated that blocking HTR2C with pizotifen might inhibit the molecular mechanisms which follow EMT induction. We first investigated the expressions of epithelial and mesenchymal markers in pizotifen-treated MDA-MB-231 cells since the activation of an EMT program needs to be transient and reversible, and transition from a fully mesenchymal phenotype to a epithelial-mesenchymal hybrid state or a fully epithelial phenotype is associated with malignant phenotypes (*Kröger et al., 2019*). IF and FACS analyses revealed 20% of pizotifen-treated MDA-MB-231 cells restored E-cadherin expression. Also, western blotting analysis demonstrated that 4T1 primary tumors from pizotifen-treated mice has elevated E-cadherin expression compared with tumors from vehicle-treated mice (*Figure 5A–C* and *Figure 5—figure supplement 1*). However, mesenchymal markers did not change between vehicle and pizotifen-treated MDA-MB-231 cells (data not shown). We further analyzed E-cadherin-positive (E-cad[+]) cells in pizotifen-treated MDA-MB-231 cells. The E-cad[+] cells showed elevated expressions of epithelial markers KRT14 and KRT19; and decreased expression of mesenchymal makers vimentin, MMP1, MMP3, and S100A4. Recent research reports that an EMT program needs to be transient and reversible and that a mesenchymal phenotype in cancer cells is achieved by constitutive ectopic expression of ZEB1. In accordance with the research, the E-cad[+] cells and 4T1 primary tumors from

**Table 7.** Effects of HTR2C overexpression on metastatic dissemination of MCF7 cells in zebrafish xenografted models.

Related to *Figure 4E*. The numbers and frequencies of the fish showing the dissemination patterns in the zebrafish that were inoculated with MCF7 cells expressing either vector control (VC) or HTR2C were indicated. The fish showed both patterns of dissemination were redundantly counted in this analysis.

|  |  | Experiment _#1 | Experiment _#2 | Average of experiments |
|---|---|---|---|---|
| VC | Non-dissemination | 46.15% n1 = 6/13 | 40% n2 = 4/10 | 43.07% ± 4.35% |
|  | Head | 46.15% n1 = 6/13 | 20% n2 = 2/10 | 33.07% ± 18.49% |
|  | Trunk | 0% n1 = 0/13 | 0% n2 = 0/10 | 0% |
|  | End-tail | 53.84% n1 = 7/13 | 60% n2 = 6/10 | 56.92% ± 4.35% |
| HTR2C | Non-dissemination | 0% n1 = 0/14 | 0% n2 = 0/15 | 0% |
|  | Head | 100% n1 = 14/14 | 93.33% n2 = 14/15 | 96.66% ± 4.71% |
|  | Trunk | 64.28% n1 = 9/14 | 73.33% n2 = 11/15 | 68.80% ± 6.39% |
|  | End-tail | 85.71% n1 = 12/14 | 93.33% n2 = 14/15 | 89.52% ± 5.38% |

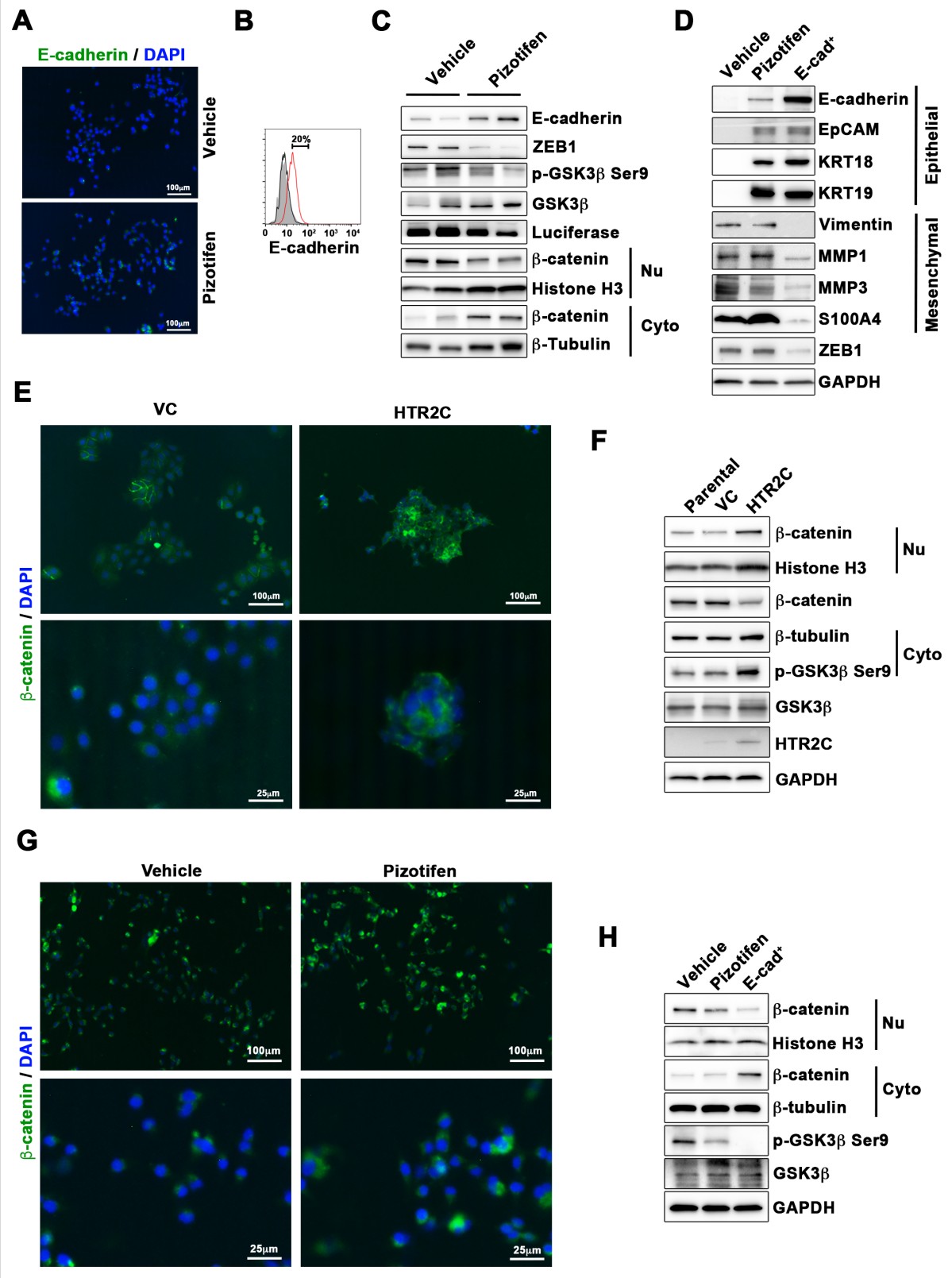

**Figure 5.** Pizotifen restored mesenchymal-like traits of MDA-MB-231 cells into epithelial traits through blocking nuclear accumulation of β-catenin. (**A**) Immunofluorescence (IF) staining of E-cadherin in either vehicle- or pizotifen-treated MDA-MB-231 cells. (**B**) Surface expression of E-cadherin in either vehicle (black)- or pizotifen (red)-treated MDA-MB-231 cells by FACS analysis. Non-stained controls are shown in gray. (**C**) Protein expressions levels of E-cadherin, ZEB1, and β-catenin in the cytoplasm and nucleus of 4T1 primary tumors from either vehicle- or pizotifen-treated mice are shown;

*Figure 5 continued on next page*

*Figure 5 continued*

Luciferase, histone H3, and β-tubulin are used as loading control for whole cell, nuclear, or cytoplasmic lysate, respectively. (**D**) Protein expression levels of epithelial and mesenchymal markers and ZEB1 in either vehicle- or pizotifen-treated MDA-MB-231 cells or E-cadherin-positive (E-cad⁺) cells in pizotifen-treated MDA-MB-231 cells are shown. (**E**) IF staining of β-catenin in the MCF7 cells expressing either vector control (top left, bottom left) or HTR2C (top right, bottom right). (**F**) Expressions of β-catenin in the cytoplasm and nucleus of MCF7 cells. (**G**) IF staining of β-catenin in either vehicle (top left, bottom left) or pizotifen-treated MDA-MB-231 cells (top right, bottom right). (**H**) Expressions of β-catenin in the cytoplasm and nucleus of MDA-MB-231 cells and the E-cad⁺ cells.

The online version of this article includes the following figure supplement(s) for figure 5:

**Figure supplement 1.** Quantification analyses of western blotting bands in *Figure 5C*.

**Figure supplement 2.** Quantification analyses of western blotting bands in *Figure 5D*.

**Figure supplement 3.** Quantification analyses of western blotting bands in *Figure 5F*.

**Figure supplement 4.** Expression of Snail and Twist1 was examined by western blotting in the MCF7 cells (left); GAPDH loading control is shown (bottom).

**Figure supplement 5.** Quantification analyses of western blotting bands in *Figure 5H*.

pizotifen-treated mice had decreased ZEB1 expression compared with vehicle-treated cells and tumors from vehicle-treated mice (*Figure 5D* and *Figure 5—figure supplement 2*). In contrast, HTR2C-expressing MCF7 and HuMEC cells expressed ZEB1 but not vehicle control MCF7 and HuMEC cells (*Figure 4C* and *Figure 5—figure supplement 3*). HTR2C-expressing MCF7 cells expressed not only ZEB1 but also Twist1 and Snail. In contrast, pizotifen-treated MDA-MB-231 cells showed decreased expression of ZEB1 and Twist1 compared with that in vehicle-treated cells. Furthermore, in the primary tumors of pizotifen-treated mice, only ZEB1 expression was decreased compared with those of vehicle-treated mice. These results indicate that HTR2C-mediated signaling induced EMT through up-regulation of ZEB1 and blocking HTR2C with pizotifen induced mesenchymal-to-epithelial transition through down-regulation of ZEB1 (*Figure 5—figure supplement 4*).

We further investigated the mechanism of action of how blocking HTR2C with pizotifen induced down-regulation of ZEB1. In embryogenesis, serotonin-mediated signaling is required for Wnt-dependent specification of the superficial mesoderm during gastrulation (*Beyer et al., 2012*). Wnt signaling plays critical role in inducing EMT. In cancer cells, overexpression of HTR1D is associated with Wnt signaling (*Sui et al., 2015*; *Zhan et al., 2017*). This evidence led to a hypothesis that HTR2C-mediated signaling might turn on transcriptional activity of β-catenin and that might induce up-regulation of EMT-TFs. IF analyses revealed β-catenin was accumulated in the nucleus of HTR2C-expressing MCF7 cells but it was located in the cytoplasm of vector control-expressing cells (*Figure 5E*). Nuclear accumulation of β-catenin in HTR2C-expressing MCF7 cells was confirmed by western blot (*Figure 5F* and *Figure 5—figure supplement 2*). In contrast, pizotifen-treated MDA-MB-231 cells showed β-catenin located in the cytoplasm of the cells. Vehicle-treated cells showed that β-catenin accumulated in the nucleus of the cells. (*Figure 5G*), and western blotting analysis confirmed that it was located in the cytoplasm of pizotifen-treated MDA-MB-231 cells (*Figure 5H* and *Figure 5—figure supplement 5*). Furthermore, immunohistochemistry and western blotting analyses showed that β-catenin accumulated in the nucleus, and phospho-GSKβ and ZEB1 expression were decreased in 4T1 primary tumors from pizotifen-treated mice compared with vehicle-treated mice (*Figure 5C* and *Figure 5—figure supplement 1*). These results indicated that HTR2C would regulate transcriptional activity of β-catenin and pizotifen could inhibit it.

Taken together, we conclude that blocking HTR2C with pizotifen restored epithelial properties to metastatic cells (MDA-MB-231 and 4T1 cells) through a decrease of transcriptional activity of β-catenin and that suppressed metastatic progression of the cells.

## Discussion

Reducing or eliminating mortality associated with metastatic disease is a key goal of medical oncology, but few models exist that allow for rapid, effective screening of novel compounds that target the metastatic dissemination of cancer cells. Based on accumulated evidence that at least 50 genes play an essential role in governing both metastasis and gastrulation progression (*Table 1*), we hypothesized that small molecule inhibitors that interrupt gastrulation of zebrafish embryos might suppress

metastatic progression of human cancer cells. We created a unique screening concept utilizing gastrulation of zebrafish embryos to test the hypothesis. Our results clearly confirmed our hypothesis: 25.6% (20/76 drugs) of epiboly-interrupting drugs could also suppress cell motility and invasion of highly metastatic human cell lines in vitro. In particular, pizotifen, which is an antagonist for serotonin receptor 2C and one of the epiboly-interrupting drugs, could suppress metastasis in a mouse model (*Figure 3A–E*). Thus, this screen could offer a novel platform for discovery of anti-metastasis drugs.

Among the 20 drugs which suppressed both epiboly progression and cell motility and invasion of MDA-MB-231 cells, hexachlorophene and troglitazone showed the strongest effect on suppressing cell motility and invasion of MDA-MB-231 cells. However, the drug could not suppress cell motility and invasion of other highly metastatic human cancer cell lines: MDA-MB-435 and PC3. With the exception of pizotifen and S(-)eticlopride hydrochloride, the remaining 18 drugs could not show the suppressor effect on more than three highly metastatic human cancer cell lines. These results indicate that the strength of interrupting effect of a drug on epiboly progression is not proportional to the strength of suppressing effect of the drug on metastasis.

We have provided the first evidence that HTR2C, which is a primary target of pizotifen, induced EMT and promoted metastatic dissemination of cancer cells (*Figure 4A–E*). Clinical data shows that HTR2C expression is correlated with tumor stage of breast cancer patients and is higher in metastatic and Her2/neu-overexpressing tumors (*Pai et al., 2009*). That would support our finding.

Pharmacological inhibition of DRD2 with S(-)eticlopride hydrochloride suppressed cell invasion and migration of multiple human cancer cell lines in vitro. However, overexpression of DRD2 could not induce EMT on MCF7 cells. Therefore, we stopped focusing on DRD2 and S(-)eticlopride hydrochloride.

There are at least two advantages to the screen described herein. One is that the screen can easily be converted to a chemical genetic screening platform. Indeed, our screen succeeded to identify HTR2C as an EMT inducer (*Figure 4A–E*). In this research, 1280 FDA approval drugs were screened, this is less than a few percent of all of druggable targets (approximately 100 targets) in the human proteome in the body. If chemical genetic screening using specific inhibitor libraries were conducted, more genes that contribute to metastasis and gastrulation could be identified. The second advantage is that the screen enables one researcher to test 100 drugs in 5 hr with using optical microscopy, drugs, and zebrafish embryos. That indicates this screen is not only highly efficient, low-cost, and low-labor but also enables researchers who do not have high-throughput screening instruments to conduct drug screening for anti-metastasis drugs.

## Materials and methods

**Key resources table**

| Reagent type (species) or resource | Designation | Source or reference | Identifiers | Additional information |
|---|---|---|---|---|
| Strain, strain background (Zebrafish) | AB line | National University of Singapore | | |
| Strain, strain background (Zebrafish) | *Tg (kdrl:eGFP)* zebrafish | Provided by Dr Stainier | | |
| Strain, strain background (*Mus musculus*) | BALB/c | Charles River Laboratories | | |
| Cell line (*Homo sapiens*) | MDA-MB-231 | ATCC | HTB-26 | |
| Cell line (*Homo sapiens*) | MCF7 | ATCC | HTB-22 | |
| Cell line (*Homo sapiens*) | MDA-MB-435 | ATCC | HTB-129 | |
| Cell line (*Homo sapiens*) | MIA-PaCa2 | ATCC | CRM-CRL-1420 | |
| Cell line (*Homo sapiens*) | PC3 | ATCC | CRL-3471 | |
| Cell line (*Homo sapiens*) | SW620 | ATCC | CCL-227 | |
| Cell line (*Homo sapiens*) | PC9 | RIKEN BRC | RCB0446 | |
| Cell line (*Homo sapiens*) | HaCaT | CLI | 300493 | |

*Continued on next page*

*Continued*

| Reagent type (species) or resource | Designation | Source or reference | Identifiers | Additional information |
|---|---|---|---|---|
| Cell line (BALB/c Mus) | 4T1-12B | Provided from Dr Gary Sahagian | | |
| Antibody | PRMT1 (A33) (Rabbit polyclonal) | Cell Signaling Technology | Cat#_2449 | WB (1:1000) |
| Antibody | CYP11A1 (D8F4F) (Rabbit polyclonal) | Cell Signaling Technology | Cat#_14217 | WB (1:1000) |
| Antibody | E-cadherin (4A2) (Mouse monoclonal) | Cell Signaling Technology | Cat#_14472 | WB (1:1000) IF (1:100) |
| Antibody | EpCAM (VU1D9) (Mouse monoclonal) | Cell Signaling Technology | Cat#_2929 | WB (1:1000) IF (1:100) |
| Antibody | Vimentin (D21H3) (Rabbit polyclonal) | Cell Signaling Technology | Cat#_5741 | WB (1:1000) IF (1:100) |
| Antibody | N-cadherin (D4R1H) (Rabbit polyclonal) | Cell Signaling Technology | Cat#_13116 | WB (1:1000) IF (1:100) |
| Antibody | ZEB1 (D80D3) (Rabbit polyclonal) | Cell Signaling Technology | Cat#_3396 | WB (1:1000) |
| Antibody | Histone H3 (D1H2) (Rabbit polyclonal) | Cell Signaling Technology | Cat#_4499 | WB (1:1000) |
| Antibody | β-Tubulin (9F3) (Rabbit polyclonal) | Cell Signaling Technology | Cat#_2128 | WB (1:1000) |
| Antibody | GAPDH (14C10) (Rabbit polyclonal) | Cell Signaling Technology | Cat#_2118 | WB (1:1000) |
| Antibody | HTR2C (ab133570) (Rabbit polyclonal) | Abcam | Cat#_ab133570 | WB (1:1000) |
| Antibody | DRD2 (ab85367) (Rabbit polyclonal) | Abcam | Cat#_ab85367 | WB (1:1000) |
| Antibody | Phospho-GSK3β (Ser9) (F-2) (Mouse monoclonal) | Santa Cruz Biotechnology | Cat#_sc-373800 | WB (1:1000) |
| Antibody | GSK3β (1F7) (Mouse monoclonal) | Santa Cruz Biotechnology | Cat#_sc-53931 | WB (1:1000) |
| Antibody | KRT18 (DC-10) (Mouse monoclonal) | Santa Cruz Biotechnology | Cat#_sc-6259 | WB (1:1000) |
| Antibody | KRT19 (A53-B/A2) (Mouse monoclonal) | Santa Cruz Biotechnology | Cat#_sc-6278 | WB (1:1000) |
| Antibody | MMP1 (3B6) (Mouse monoclonal) | Santa Cruz Biotechnology | Cat#_sc-21731 | WB (1:1000) |
| Antibody | MMP2 (8B4) (Mouse monoclonal) | Santa Cruz Biotechnology | Cat#_sc-13595 | WB (1:1000) |
| Antibody | S100A4 (A-7) (Mouse monoclonal) | Santa Cruz Biotechnology | Cat#_sc-377059 | WB (1:1000) |
| Antibody | Luciferase (C-12) (Mouse monoclonal) | Santa Cruz Biotechnology | Cat#_sc-74548 | WB (1:1000) |
| Antibody | ki67 (ki-67) (Mouse monoclonal) | Santa Cruz Biotechnology | Cat#_sc-23900 | WB (1:1000) |
| Antibody | β-Catenin (E-5) (Mouse monoclonal) | Santa Cruz Biotechnology | Cat#_sc-7963 | WB (1:1000) IF (1:100) |
| Antibody | FITC-conjugated E-cadherin antibody (67A4) | Biolegend | Cat#_324104 | FACS (1:100) |

*Continued on next page*

*Continued*

| Reagent type (species) or resource | Designation | Source or reference | Identifiers | Additional information |
|---|---|---|---|---|
| Antibody | Anti-mouse anti-rabbit immunoglobulin G (IgG) antibodies conjugated to Alexa Fluor 488 | Life Technologies | A-11029 | IF (1:100) |
| Antibody | Anti-goat anti-rabbit immunoglobulin G (IgG) antibodies conjugated to Alexa Fluor 488 | Life Technologies | A-11034 | IF (1:100) |
| Recombinant DNA reagent | pLVX-shRNA1 | Clontech | Cat#_ 632,177 | |
| Recombinant DNA reagent | pCDH-CMV-MCS-EF1α-Hygro | System Biosciences | Cat#_CD515B-1 | Gene expression vector |
| Recombinant DNA reagent | pMDLg/pRRE | Addgene | Addgene Plasmid #12251 RRID:Addgene_12251 | Lentivirus packaging vector |
| Recombinant DNA reagent | pRSV-rev | Addgene | Addgene Plasmid #12253 RRID:Addgene_12253 | Lentivirus packaging vector |
| Recombinant DNA reagent | pMD2.G | Addgene | Addgene Plasmid #12259 RRID:Addgene_12259 | Lentivirus packaging vector |
| Recombinant DNA reagent | Providing pCMV-h5TH2C-VSV | Provided from Dr Herrick | | |
| Chemical compound, drug | FDA-approved chemical libraries | Prestwick Chemical | | |
| Chemical compound, drug | Pizotifen | Santa Cruz Biotechnology | Cat#_sc-201143 | |
| Chemical compound, drug | S(-)Eticlopride hydrochloride | Santa Cruz Biotechnology | Cat#_E101 | |
| Software, algorithm | GraphPad Prism7 | GraphPad Software Inc | RRID:SCR_002798 | Data analysis |
| Software, algorithm | FlowJo | BD Biosciences | RRID:SCR_008520 | FACS data analysis |

## Zebrafish embryo screening

Zebrafish embryos at two-cell stage were collected at 20 min after their fertilization. Each drug was added to a well of a 24-well plate containing approximately 20 zebrafish embryos per well in either 10 or 50 µM final concentration when the embryos reached the sphere stage. Chemical treatment was initiated at 4 hpf and approximately 20 embryos were treated with two different concentrations for each compound tested. The treatment was ended at 9 hpf when vehicle- (DMSO) treated embryos as control reach 80–90% completion of the epiboly stage. The compounds which induced delay (<50% epiboly) in epiboly were selected as hit compounds for in vitro testing using highly metastatic human cancer cell lines. The study protocol was approved by the Institutional Animal Care and Use Committee of the National University of Singapore (protocol number: R16-1068).

## Reagents

FDA, EMA, and other agencies-approved chemical libraries were purchased from Prestwick Chemical (Illkirch, France). Pizotifen (sc-201143) and S(-)eticlopride hydrochloride (E101) were purchased from Santa Cruz (Dallas, TX) and Sigma-Aldrich (St Louis, MO).

## Cell culture and cell viability assay

MCF7, MDA-MB-231, MDA-MB-435, MIA-PaCa2, PC3, SW620, PC9, and HaCaT cells were obtained from American Type Culture Collection (ATCC, Manassas, VA). Luciferase-expressing 4T1 (4T1-12B) cells were provided from Dr Gary Sahagian (Tufts University, Boston, MA). All culture methods followed the supplier's instruction. Cell viability assay was performed as previously described (*Nakayama et al., 2020*). PCR-based mycoplasma testing on these cells was performed once in 4 months.

## Plasmid

A DNA fragment coding for HTR2C was amplified by PCR with primers containing restriction enzyme recognition sequences. The HTR2C coding fragment was amplified from hsp70l:mCherry-T2A-CreERT2 plasmid (*Huang et al., 2012*).

## Immunoblotting

Western blotting was performed as described previously (*Nakayama et al., 2020*). Raw data of images of western blotting analyses are uploaded as source data for western. Anti-PRMT1 (A33), anti-CYP11A1 (D8F4F), anti-E-cadherin (4A2), anti-EpCAM (VU1D9), anti-vimentin (D21H3), anti-N-cadherin (D4R1H), anti-ZEB1 (D80D3), anti-histone H3 (D1H2), anti-β-tubulin (9F3), and anti-GAPDH (14C10) antibodies were purchased from Cell Signaling Technology (Danvers, MA). Anti-HTR2C (ab133570) and anti-DRD2 (ab85367) antibodies were purchased form Abcam (Cambridge, UK). Anti-phospho-GSK3β (Ser9) (F-2), anti-GSK3β (1F7), anti-KRT18 (DC-10), anti-KRT19 (A53-B/A2), anti-MMP1 (3B6), anti-MMP2 (8B4), anti-S100A4 (A-7), anti-luciferase (C-12), anti-ki67 (ki-67), and anti-β-catenin (E-5) antibodies were purchased from Santa Cruz Biotechnology (Dallas, TX).

## Flow cytometry

Cells were stained with FITC-conjugated E-cadherin antibody (67A4, Biolegend, San Diego, CA). Flow cytometry was performed as described (*Nakayama et al., 2009*) and analyzed with FlowJo software (TreeStar, Ashland, OR).

## shRNA-mediated gene knockdown

The shRNA-expressing lentivirus vectors were constructed using pLVX-shRNA1 vector (632177, TAKARA Bio, Shiga, Japan). PRMT1-shRNA_#3-targeting sequence is GTGTTCCAGTATCTCTGATT A; PRMT1-shRNA_#4-targeting sequence is TTGACTCCTACGCACACTTTG. CYP11A1-shRNA_#4-targeting sequence is GCGATTCATTGATGCCATCTA; CYP11A1-shRNA_#4-targeting sequence is GAAATCCAACACCTCAGCGAT. Human HTR2C-shRNA-targeting sequence is TCATGCACCTCT GCGCTATAT. Mouse HTR2C-shRNA-targeting sequence is CTTCATACCGCTGACGATTAT. LacZ-shRNA-targeting sequence is CTACACAAATCAGCGATT.

## Immunofluorescence

IF microscopy assay was performed as previously described (*Nakayama et al., 2020*). Goat anti-mouse and goat anti-rabbit immunoglobulin G (IgG) antibodies conjugated to Alexa Fluor 488 (A-11029 and A-11034, Life Technologies, Carlsbad, CA) and diluted at 1:100 were used. Nuclei were visualized by the addition of 2 µg/ml of 4',6-diamidino-2-phenylindole (DAPI) (62248, Thermo Fisher, Waltham, MA) and photographed at 100× magnification by a fluorescent microscope BZ-X700 (KEYENCE, Osaka, Japan).

## Boyden chamber cell motility and invasion assay

These assays were performed as previously described (*Nakayama et al., 2020*). In Boyden chamber assay, either $3 \times 10^5$ MDA-MB-231, $1 \times 10^6$ MDA-MB-435 or $5 \times 10^5$ PC9 cells were applied to each well in the upper chamber.

## Zebrafish xenotransplantation model

*Tg(kdrl:eGFP)* zebrafish was provided by Dr Stainier (Max Planck Institute for Heart and Lung Research). Embryos that were derived from the line were maintained in E3 medium containing 200 µM 1-phenyl-2-thiourea (P7629, Sigma-Aldrich, St Louis, MO). Approximately 100–400 RFP-labelled MBA-MB-231 or MIA-PaCa2 cells were injected into the duct of Cuvier of the zebrafish at 2 dpf. The fish were randomly assigned to two groups. One group was maintained in the presence of pizotifen-containing E3 medium and the other group was maintained in vehicle-containing E3 medium.

## Spontaneous metastasis mouse model

4T1-12B cells ($2 \times 10^4$) were injected into the #4 MFP while the mice were anesthetized. To monitor tumor growth and metastases, mice were imaged biweekly by IVIS Imaging System (ParkinElmer, Waltham, MA). The primary tumor was resected 10 days after inoculation. D-Luciferin Potassium Salt

(LUCK-100) was purchased from GoldBio (St Louis, MO). The study protocol (protocol number: BRC IACUC #110612) was approved by A*STAR (Agency for Science, Technology and Research, Singapore).

## Gene set enrichment analysis

Gene expression profiles obtained from zebrafish embryos at either 50%-epiboly, shield, or 75%-epiboly stage were analyzed based on the hallmark gene sets derived from the Molecular Signatures Database (MSigDB) (*Subramanian et al., 2005*; *Liberzon et al., 2015*). The zebrafish transcriptomic data was sourced from *White et al., 2017*. Gene sets that were significantly enriched (FDR < 0.25) were presented with the normalized enrichment score (NES) and nominal p value. Source data files for analysis of either gene expression and enriched pathways are uploaded as GSEA *Source data 1* and *2*, respectively.

## Histological analysis

All OCT-embedded primary tumors, lungs, and livers of mice from the spontaneous metastasis 4T1 model were sectioned on a cryostat. Eight micron sections were taken at 500 μm intervals through the entirety of the livers and lungs. Sections were subsequently stained with hematoxylin and eosin. Metastatic lesions were counted under a microscope in each section for both lungs and livers.

## Statistics

Data were analyzed by Student's t test; $p < 0.05$ was considered significant.

## Acknowledgements

We sincerely appreciate Dr Joshua Collins (NIH/NIDCR) and Dr Shimada (Mie University) for helping this research. We thank Dr Herrick (Albany Medical College) for providing pCMV-h5TH2C-VSV with us. This study was funded by grants from National Medical Research Council of Singapore (R-154000547511) and Ministry of Education of Singapore (R-154000A23112) to ZG.

## Additional information

### Funding

| Funder | Grant reference number | Author |
| --- | --- | --- |
| National Medical Research Council | R-154000547511 | Zhiyuan Gong |
| Ministry of Education - Singapore | R-154000A23112 | Zhiyuan Gong |

The funders had no role in study design, data collection and interpretation, or the decision to submit the work for publication.

### Author contributions

Joji Nakayama, Conceptualization, Data curation, Formal analysis, Investigation, Methodology, Supervision, Validation, Visualization, Writing - original draft, Writing - review and editing; Lora Tan, Formal analysis, Investigation, Validation, Visualization; Yan Li, Data curation, Investigation; Boon Cher Goh, Hideki Makinoshima, Funding acquisition, Project administration, Resources; Shu Wang, Funding acquisition, Resources; Zhiyuan Gong, Funding acquisition, Project administration, Resources, Supervision

### Author ORCIDs

Joji Nakayama (ORCID) https://orcid.org/0000-0003-1077-140X

### Ethics

The study protocol using zebrafish was approved by the Institutional Animal Care and Use Committee of the National University of Singapore (protocol number: R16-1068). The study protocol using mice (protocol number: BRC IACUC #110612) was approved by A*STAR (Agency for Science, Technology and Research, Singapore).

Decision letter and Author response

Decision letter https://doi.org/10.7554/eLife.70151.sa1

Author response https://doi.org/10.7554/eLife.70151.sa2

## Additional files

### Supplementary files

• Transparent reporting form

• Source data 1. GSEA analysis of zebrafish embryos at either 50%-epiboly, shield or 75%-epiboly stage.

• Source data 2. Enriched pathways of zebrafish embryos at either 50%-epiboly, shield or 75%-epiboly stage.

• Source data 3. Raw data of western-blotting analysis.

• Source data 4. Raw data of western-blotting analysis with legends.

### Data availability

All data generated or analysed during this study are included in the manuscript and supporting files.

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
