## [Editor Report]

We are so impressed with this new and ambitious concept for chemical screening using zebrafish embryos to find a novel anti-metastasis drug, Pizotifen. We hope many researchers will use this screening system for anti-cancer drug discovery.

---

## [Decision Letter]

**Decision letter after peer review:**

Thank you for submitting your article "A chemical screen based on an interruption of zebrafish gastrulation identifies the HTR2C inhibitor Pizotifen as a suppressor of EMT-mediated metastasis" for consideration by *eLife*. Your article has been reviewed by 3 peer reviewers, and the evaluation has been overseen by a Reviewing Editor and Didier Stainier as the Senior Editor. The reviewers have opted to remain anonymous.

Three reviewers basically agree that your paper should be published in *eLife*. And I also think it is a very unique and interesting paper suitable for *eLife*. However, two reviewers think that additional data are necessary to strengthen your conclusions. I would be very grateful if you could perform these experiments in order to publish your work with more impact.

Comprehensive analysis using sequencing data is needed to support the concept that human cancer metastasis mimic/recapitulate zebrafish gastrulation. In addition, Pizotifen's functional part needs additional experiments and more supportive data. Details are in the recommendations to the authors.

*Reviewer #1 (Recommendations for the authors):*

The authors need to provide a detailed catalog numbers of the reagents used in this study.

*Reviewer #2 (Recommendations for the authors):*

1. It is not clear how the authors narrowed down the list from 66 hits to 22 compounds and further why Pizotifen was selected for further study. Can the authors provide additional information or data to justify exclusion of hits. Based on the data the compounds that effect dopamine based signaling would have been a more obvious pathway to study (4/22 hits).

2. The selection of human cancer cell lines to validate the findings are not clear. Is HTR2C relevant only to the subset of cancer cell lines studied. Analyses of TCGA or other data sets to show relevance of HTR2C gene expression and survival and or metastasis would greatly strengthen the rationale for picking these cell lines. Is HTR2C relevant for metastasis in all cancers or only in breast cancer. Also, it appears in figure 2B that MCF non metastatic breast cancer cells express higher levels of HTR2C than metastatic skin (MDA-MB-435) and lung (PC9) cells.

3. Some of the data provided does not support the conclusions drawn. For example, In figure 3 C. The luciferase images clearly show that Pizotifen treated mice have smaller tumors while the data in 3 A and B do not support this data. The measurements of tumor volume appear to be taken too early to actually be reproducible, also the data for proliferation and apoptosis was not shown. The authors should provide more data to justify the conclusions drawn.

4. Figure 5 A, 5B, 5G. The effects shown in the texts are not clearly visible. Can the authors provide higher magnification (40X) images to show differences noted in the text.

5. In figure 5D or 5H the effect on Pizotifen on ZEB1 or β-catenin reduction is not observed. Does this suggest this compound is effecting EMT and metastasis by other mechanisms?

6. It is not clear if HTR2C selectively only inhibits ZEB1 mediated metastases or are its effects more general i.e. its expression also effects other EMT regulators for example SNAIL, SLUG and or TWIST expression. Showing the effect of Pizotifen/ HTR2C on other EMT regulators especially the ones above is recommended.

*Reviewer #3 (Recommendations for the authors):*

1. Line 142: Mentioned DRD2 but did not elaborate on it and explain why it wasn't chosen as a target. In addition, please provide the rationale, why authors forgo other drugs, including the two with the strongest effect, Hexachloroghene and Troglitazone, to focus on Pizotifen.

2. Line 307, please indicate: in which cancer cells is overexpression of HTR1D associated with Wnt-signaling that enables induction of EMT?

3. Figure 5E and 5G: the figures do not show convincing nuclear accumulation of ß-catenin. Please show zoom in insert figures.

[Editors' note: further revisions were suggested prior to acceptance, as described below.]

Thank you for resubmitting your work entitled "A chemical screen based on an interruption of zebrafish gastrulation identifies the HTR2C inhibitor Pizotifen as a suppressor of EMT-mediated metastasis" for further consideration by *eLife*. Your revised article has been reviewed by 3 peer reviewers and the evaluation has been overseen by Didier Stainier as the Senior Editor, and a Reviewing Editor.

We are impressed with Dr. Nakayama's manuscript with a new and ambitious concept for the chemical screening for cancer metastasis using zebrafish embryos. However, two reviewers have requested that the authors clarify the mechanisms related to EMT. I also agree with their opinions because EMT is one of the most important factors for cancer metastasis and is well-studied already. It is very much expected by other researchers and would strengthen your story.

*Reviewer #1 (Recommendations for the authors):*

EMT is just one of the numerous steps of both gastrulation and tumor metastasis, as they mentioned that the transcriptomic data for zebrafish embryo development at the epiboly/gastrulation stage are based on the whole embryos which include all other activities and are not specific to EMT. As far as they observe phenotypic changes of whole embryo and focus on the similarity between gastrulation and metastasis in the 1st screening, what they should analyze in RNA-seq data is "global similarity" but not EMT. While they mentioned that this point is not really an objective of this study, I still feel this is one of the most important point that can be resolved, although they eventually find a drug that affects EMT.

Insufficiency of mechanistic parts has not been addressed.

Overall, I didn't feel in the revised manuscript that the authors have addressed many of my concerns, and recommend to resubmit to the other journal.

*Reviewer #2 (Recommendations for the authors):*

The authors have addressed most of the major comments except for Major comment 6. The authors provide no data (only text below) showing expression of SNAIL, SLUG or TWIST in MCF7 cells with and without Pizotifen, or any data from the mouse xenograft experiments. Since the authors emphasize the role of ZEB1 showing that other EMT factors are not affected or not consistently affected would help. At a minimum the authors should add their statement below to the text.

6. It is not clear if HTR2C selectively only inhibits ZEB1 mediated metastases are its effects more general i.e. its expression also effects other EMT regulators for example SNAIL, SLUG and or TWIST expression. Showing the effect of Pizotifen/ HTR2C on other EMT regulators especially the ones above is recommended.

Author response: HTR2C-expressing MCF7 cells expressed not only ZEB1 but also Twist1 and SNAIL. In contrast, Pizotifen-treated MDA-MB-231 cells showed decreased expression of ZEB1 and Twist1 compared with that in vehicle-treated the cells. In the primary tumors of Pizotifen-treated mice, only ZEB1 expression was decreased compared with those of vehicle-treated mice."

---

## [Author Response]

Reviewer #1 (Recommendations for the authors):The authors need to provide a detailed catalog numbers of the reagents used in this study.

I added catalog numbers of the reagents used in this study.

Reviewer #2 (Recommendations for the authors):1. It is not clear how the authors narrowed down the list from 66 hits to 22 compounds and further why Pizotifen was selected for further study. Can the authors provide additional information or data to justify exclusion of hits. Based on the data the compounds that effect dopamine based signaling would have been a more obvious pathway to study (4/22 hits).

Zebrafish embryo screen identified 78 drugs as epiboly-interrupting drugs. Sixteen of the 78 drugs strongly affected cell viability at concentrations less than 1µM and were not used in the cell motility experiments. The remaining 62 drugs were assayed in Boyden chamber motility experiments. Twenty of the 62 drugs enabled to inhibit cell motility and invasion of MDAMB-231 cells without effecting cell viability and proceeded to next examination. The 42 drugs were passed over.

Inhibiting Dopamine receptor D2 (DRD2) suppressed metastatic dissemination of human cancer cells in a zebrafish xenografted model. However, overexpression of DRD2 could not induce EMT on MCF7 cells. Therefore, we have stop focusing DRD2. We incorporated this point into discussion part, line 777 to 781.

2. The selection of human cancer cell lines to validate the findings are not clear. Is HTR2C relevant only to the subset of cancer cell lines studied. Analyses of TCGA or other data sets to show relevance of HTR2C gene expression and survival and or metastasis would greatly strengthen the rationale for picking these cell lines.

According to cancer cell line encyclopedia, *HTR2C* mRNA expression is observed in cell lines which are established from embryo carcinoma, ovarian cancer, brain cancer, neuroblastoma, and myeloma. Ectopic expression of HTR2C is observed in not only the human cancer cell lines used in this study but also other cancer cell lines.

The reason why we select the human cancer cell lines used in this study, is that they are commonly used in an experimental study of metastasis and show metastasis in high frequency when they are xenografted into mice.

Is HTR2C relevant for metastasis in all cancers or only in breast cancer.

Pai et al., Breast Cancer Research 2009 shows that HTR2C expression is correlated with tumor stage of breast cancer patients and is higher in metastatic and Her2/ neu-overexpressing tumors (Vaibhav P Pai et al., 2009). We incorporate this point to main text in line 270-272.

Also, it appears in figure 2B that MCF non metastatic breast cancer cells express higher levels of HTR2C than metastatic skin (MDA-MB-435) and lung (PC9) cells.

Serotonin binds to HTR2C and activates HTR2C-mediated signaling. Biochemically, tryptophan hydroxylase 1 (TPH1) catalyzes amino acid tryptophan and yields serotonin. Serotonin in in vitro condition is provided from either an autocrine production of serotonin in cancer cells or culture media. Figure 5C clearly showed nuclear accumulation of β-catenin and p-GSK-3β were not observed in MCF7 cells. The result suggest serotonin would not exist in culture media and an autocrine production of serotonin would activate HTR2C-mediated signaling. Pai et al., revealed that MDA-MB-231 cells express TPH1 but MCF7 cells not. According to cancer cell line encyclopedia*, TPH1* mRNA expressions in MDAMB-435 and PC9 cells is approximately five and two-fold rather than that in MCF7 cells, respectively. Taken together, both of HTR2C expression and an autocrine production of serotonin by TPH1 would need to activate HTR2C-mediated signaling.

3. Some of the data provided does not support the conclusions drawn. For example, In figure 3 C. The luciferase images clearly show that Pizotifen treated mice have smaller tumors while the data in 3 A and B do not support this data. The measurements of tumor volume appear to be taken too early to actually be reproducible, also the data for proliferation and apoptosis was not shown. The authors should provide more data to justify the conclusions drawn.

4T1 tumors grow aggressively in the primary sites and the tumor sizes often exceed the limits that are allowed in most animal protocols. Therefore, in 4T1 metastasis model, the primary tumors are usually resected at day 10 post inoculation. Following the conventional protocol on the model, we resected primary tumors at day 10 post inoculation. The images of top left and top right of Figure 3C were taken before the resection. In the image of bottom left of Figure 3C, bioluminescence emissions are detected around the mammary fat pad of vehicle-treated mice but the emissions might arise from either the lymph node or unresected primary tumor.

We have added data of apoptosis status in primary tumor of either vehicle or Pizotifen-treated mice as Figure S4. Proliferation status in primary tumor of either vehicle or Pizotifen-treated mice are indicated in Figure 3B.

4. Figure 5 A, 5B, 5G. The effects shown in the texts are not clearly visible. Can the authors provide higher magnification (40X) images to show differences noted in the text.

We inserted higher magnification images in Figure 5E and 5G. By using three different methods: immunofluorescence images, FACS and western blotting, we demonstrate Pizotifen recovers E-cadherin expression on a part of MDA-MB-231 cells. Immunofluorescence images in Figure 5A point that a part of Pizotifen-treated MDA-MB-231 cells show Ecadherin expression. Higher magnification image in Figure 5A would have missed the point.

5. In figure 5D or 5H the effect on Pizotifen on ZEB1 or β-catenin reduction is not observed. Does this suggest this compound is effecting EMT and metastasis by other mechanisms?

Quantification analyses of western blotting bands in Figure 5H reveals that b-catenin accumulation in the nucleus of Pizotifen-treated MDA-MB-231 cells decreased compared with those of vehicle-treated ones. Similar analyses in Figure 5D shows ZEB1 expression of Pizotifen-treated MDA-MB-231 cells slightly decreased compared with those of vehicle-treated ones. ZEB1 expression was significantly decreased in E-cadherin positive fraction of Pizotifen-treated MDA-MB-231 cells compared with ZEB1 expression in vehicle-treated ones. Therefore, we conclude Pizotifen affects ZEB1 expression.

6. It is not clear if HTR2C selectively only inhibits ZEB1 mediated metastases or are its effects more general i.e. its expression also effects other EMT regulators for example SNAIL, SLUG and or TWIST expression. Showing the effect of Pizotifen/ HTR2C on other EMT regulators especially the ones above is recommended.

HTR2C-expressing MCF7 cells expressed not only ZEB1 but also Twist1 and SNAIL. In contrast, Pizotifen-treated MDA-MB-231 cells showed decreased expression of ZEB1 and Twist1 compared with that in vehicle-treated the cells. In the primary tumors of Pizotifen-treated mice, only ZEB1 expression was decreased compared with those of vehicle-treated mice.

Reviewer #3 (Recommendations for the authors):1. Line 142: Mentioned DRD2 but did not elaborate on it and explain why it wasn't chosen as a target.

We incorporated this point to discussion part, line 777-781.

Inhibiting Dopamine receptor D2 (DRD2) suppressed cell invasion and migration of multiple human cancer cell lines in vitro However, overexpression of DRD2 could not induce EMT on MCF7 cells. In contrast, pharmacologic and genetic inhibition of HTR2C suppressed metastatic progression; conversely, overexpression of HTR2C induced EMT. Therefore, we focus on Pizotifen.

In addition, please provide the rationale, why authors forgo other drugs, including the two with the strongest effect, Hexachloroghene and Troglitazone, to focus on Pizotifen.

We incorporated this point to discussion part, line 763-771. We have two reasons why we did not focus on Hexachloroghene. One is that Hexachloroghene strongly inhibited cell invasion and migration of MDA-MB-231 cells but did not affect cell invasion and migration of other cells: MDA-MB-436 and PC9 cells.

Second is that commercial products including Hexachloroghene killed 15 babies in the

United States and 39 babies in France in 1972. Even if we could demonstrate

Hexachloroghene has suppressor effect on metastasis, an issue of the toxic effects might abolish the new discovery. Therefore, we stopped focusing Hexachloroghene.

We have one reason why we did not focus on Troglitazone. Troglitazone has potential high liver toxicity and leads to drug-induced hepatitis. Once troglitazone was approved by FDA, it was withdrawn from the British market in December 1997, from the United States market in 2000, and from the Japanese market soon afterwards. Even if we could demonstrate Troglitazone has suppressor effect on metastasis, an issue of the toxic effects might abolish the new discovery. Therefore, we stopped focusing Troglitazone.

2. Line 307, please indicate: in which cancer cells is overexpression of HTR1D associated with Wnt-signaling that enables induction of EMT?

This paragraph is changed into "Wnt-signaling plays critical role in inducing EMT. In cancer cells, overexpression of HTR1D is associated with Wnt-signaling".

3. Figure 5E and 5G: the figures do not show convincing nuclear accumulation of ß-catenin. Please show zoom in insert figures.

We inserted higher magnification images in Figure 5E and 5G.

[Editors' note: further revisions were suggested prior to acceptance, as described below.]

Reviewer #1 (Recommendations for the authors):EMT is just one of the numerous steps of both gastrulation and tumor metastasis, as they mentioned that the transcriptomic data for zebrafish embryo development at the epiboly/gastrulation stage are based on the whole embryos which include all other activities and are not specific to EMT. As far as they observe phenotypic changes of whole embryo and focus on the similarity between gastrulation and metastasis in the 1st screening, what they should analyze in RNA-seq data is "global similarity" but not EMT. While they mentioned that this point is not really an objective of this study, I still feel this is one of the most important point that can be resolved, although they eventually find a drug that affects EMT.

We compared the genes expressed in zebrafish gastrulation with the gene related with EMT through performing gene set enrichment analysis (GSEA). The analysis revealed that 50% epiboly, shield and 75%-epiboly stage of zebrafish embryos expressed the genes which promote EMT-mediated metastasis: EMT induction, TGF-β signaling, wnt/β-catenin signaling and Notch signaling (Figure 1—figure supplemental 1). This data demonstrates that a part of genes are commonly expressed between zebrafish gastrulation and EMT-mediated metastasis of human cancer cells.

Reviewer #2 (Recommendations for the authors):The authors have addressed most of the major comments except for Major comment 6. The authors provide no data (only text below) showing expression of SNAIL, SLUG or TWIST in MCF7 cells with and without Pizotifen, or any data from the mouse xenograft experiments. Since the authors emphasize the role of ZEB1 showing that other EMT factors are not affected or not consistently affected would help. At a minimum the authors should add their statement below to the text.6. It is not clear if HTR2C selectively only inhibits ZEB1 mediated metastases are its effects more general i.e. its expression also effects other EMT regulators for example SNAIL, SLUG and or TWIST expression. Showing the effect of Pizotifen/ HTR2C on other EMT regulators especially the ones above is recommended.Author response: HTR2C-expressing MCF7 cells expressed not only ZEB1 but also Twist1 and SNAIL. In contrast, Pizotifen-treated MDA-MB-231 cells showed decreased expression of ZEB1 and Twist1 compared with that in vehicle-treated the cells. In the primary tumors of Pizotifen-treated mice, only ZEB1 expression was decreased compared with those of vehicle-treated mice."

We added following statement at line 321-328 and graphic data as Figure 5—figure supplement 4. “HTR2C-expressing MCF7 cells expressed not only ZEB1 but also Twist1 and Snail. […] These results indicate that HTR2C-mediated signaling induced EMT through up-regulation of ZEB1 and blocking HTR2C with Pizotifen induced mesenchymal to epithelial transition through downregulation of ZEB1 (Figure 5—figure supplement 4).”